# Terminal deoxynucleotidyl transferase and CD84 identify human multi-potent lymphoid progenitors

YeEun Kim [1,2], Ariel A. Calderon[1,2], Patricia Favaro[2], David R. Glass [1,2], Albert G. Tsai [2], Daniel Ho[2], Luciene Borges [2], William J. Greenleaf [3] ✉ & Sean C. Bendall [2] ✉

Lymphoid specification in human hematopoietic progenitors is not fully understood. To better associate lymphoid identity with protein-level cell features, we conduct a highly multiplexed single-cell proteomic screen on human bone marrow progenitors. This screen identifies terminal deoxynucleotidyl transferase (TdT), a specialized DNA polymerase intrinsic to VDJ recombination, broadly expressed within CD34+ progenitors prior to B/T cell emergence. While these TdT+ cells coincide with granulocyte-monocyte progenitor (GMP) immunophenotype, their accessible chromatin regions show enrichment for lymphoid-associated transcription factor (TF) motifs. TdT expression on GMPs is inversely related to the SLAM family member CD84. Prospective isolation of CD84lo GMPs demonstrates robust lymphoid potentials ex vivo, while still retaining significant myeloid differentiation capacity, akin to LMPPs. This multi-omic study identifies human bone marrow lymphoid-primed progenitors, further defining the lympho-myeloid axis in human hematopoiesis.

Our understanding of hematopoiesis has evolved dramatically in the last decade with the advances in single-cell technologies. Traditionally, hematopoiesis has been portrayed as a hierarchical system in which hematopoietic stem cells (HSCs) differentiate into oligo-potential and uni-potential progenitors in a stepwise manner. And each progenitor population with a distinct differentiation potential was identified by their expression of specific cell surface proteins, also known as 'surface markers' (Table 1). However, single-cell techniques revealed the continuous transcriptomic[1], epigenetic[2,3], and proteomic[4] landscapes of human hematopoietic stem and progenitor cells (HSPCs). In parallel, single-cell level differentiation studies demonstrated that cells within the same 'population' based on the surface markers exhibit heterogeneous differentiation potentials[5,6]. Thus, the initial surface phenotype definitions of HSPC. Cell types need updates to reflect recent findings.

In human hematopoiesis, conflicting observations in lymphoid development prompt a deeper examination of lymphoid potentials. Under the hierarchical model, lymphoid priming begins in lymphoid-primed multipotent progenitors (LMPPs), also known as

multi-lymphoid progenitors (MLPs)[7,8], and continues in the common lymphoid progenitors (CLPs)[9]. LMPPs exist in the most immature CD38lo compartment and have lymphoid and myeloid potentials but no erythro-megakaryocytic potentials[7,8]. Conceptually, CLPs are downstream of LMPPs and can generate all lymphoid lineages (T, NK, B, and pDC), but none of the other lineages[9]. A recent single-cell transcriptomic study has reported that the CD38lo HSPCs exhibit largely unstable clustering results and are highly interconnected in the nearest neighbor graph[1]. From their findings, Velten et al. suggested that the CD38lo progenitors represent very early transitory states, in which the lineage priming is beginning, rather than discrete cell types. On the other hand, the upregulation of CD10 has been associated with B cell differentiation bias and a high degree of IgH DJ rearrangement[10,11], suggesting CD10+ CLPs to be B-lineage-primed progenitors. Thus, current definitions for human HSPCs only identify the very beginning of lymphoid priming and primarily B-lineage committed cells, leaving the intermediate stages of T and NK lymphoid development largely undefined.

[1]Immunology Graduate Program, Stanford University, Stanford, CA, USA. [2]Department of Pathology, Stanford University, Stanford, CA, USA. [3]Department of Genetics, Stanford University, Stanford, CA, USA. ✉e-mail: wjg@stanford.edu; bendall@stanford.edu

**Table 1 | Human hematopoietic stem and progenitor cell population definitions**

| HSPC Cell type | Abbreviation | Lineage Potentials | Immunophenotypes |
|---|---|---|---|
| Hematopoietic stem cell | HSC | All blood-cell lineages, with self-renewal | Lin⁻CD34⁺CD38ˡᵒCD90⁺CD45RA⁻ |
| Multipotent progenitor | MPP | All blood-cell lineages, no self-renewal | Lin⁻CD34⁺CD38ˡᵒCD90⁻CD45RA⁻ |
| Lymphoid–primed multipotent progenitor | LMPP | Mo, Gr, DCs, T, B, NK, | Lin⁻CD34⁺CD38ˡᵒCD90⁻CD45RA⁻ |
| Common Myeloid Progenitor | CMP | Ery, Mk, Mo, Gr, DCs | Lin⁻CD34⁺CD38ʰⁱCD10⁻CD123ᵐᵉᵈCD45RA⁻ |
| Common lymphoid progenitor | CLP | T, B, NK, pDC | Lin⁻CD34⁺CD38ʰⁱCD10⁺ |
| Megakaryocyte-erythrocyte progenitor | MEP | Ery, Mk | Lin⁻CD34⁺CD38ʰⁱCD10⁻CD123⁻CD45RA⁻ |
| Granulocyte-monocyte progenitor | GMP | Mo, Gr, DCs | Lin⁻CD34⁺CD38ʰⁱCD10⁻CD123ᵐᵉᵈCD45RA⁺ |
| Plasmacytoid dendritic cell progenitor | pDC | pDC | Lin⁻CD34⁺CD38ʰⁱCD10⁻CD123⁺CD45RA⁺ |

*Lin* lineage markers, *Mo* monocytes, *Gr* granulocytes, *DC* dendritic cells, *pDC* plasmacytoid dendritic cells, *Ery* erythrocytes, *Mk* megakaryocytes.
Summary of the HSPC cell type definitions used in this manuscript.

Lymphoid potential has been reported outside the human LMPP or CLP compartments. Our single-cell mass cytometry study identified lymphoid-phenotype progenitors with expressions of terminal deoxynucleotidyl transferase (TdT) or IL-7Ra, but not CD10[11]. These progenitors were located in between CD38ˡᵒ progenitors and CD10⁺ cells in the B cell development pseudotime, suggesting a lymphoid-specific gene regulation activated in between LMPPs and CD10⁻ CLPs. Lin⁻CD34⁺CD38⁺CD10⁻CD45RA⁺CD62Lʰⁱ progenitor cells also exhibited full lymphoid and monocytic potentials[12]. However, the instability of CD62L in the freeze-thaw cycle[13] makes it inadequate for frozen BM samples, which are often used in research settings. Nonetheless, Kohn et al. successfully showed the existence of lymphoid progenitors that are not captured in LMPP or CLP gates. Subsequently, residual lymphoid potential was even reported within the canonical granulocyte-monocyte progenitors (GMPs)[6] in the human cord blood. Yet the phenotypic nature of the cells responsible for the lymphoid progenitor activity remained elusive. Altogether, these studies highlight the need for assessment of molecular and functional lymphoid potential that could be linked directly to cellular phenotypes within the current human HSPC rubric.

Considering the frequent conservation of mouse and human hematopoiesis[14–16], it is natural to compare the lymphopoiesis of the mice and the humans. Similar to the human CD10⁺ CLPs, the initially described murine CLPs (Lin⁻Sca-1ˡᵒcKitˡᵒIL-7R⁺Thy-1⁻)[17] were criticized for their strong B cell bias[18,19]. It is now understood that among the LSK IL-7R⁺Thy-1⁻ progenitors, Ly6D⁺ cells are B cell progenitors, and Ly6D⁻ cells are the progenitors with all lymphoid potentials[20]. Yet, the corresponding population of Ly6D⁻ CLPs remains ambiguous in humans, emphasizing the knowledge gap in the human lymphopoiesis.

In this study, we take a bottom-up, data-driven approach to identify lymphoid progenitors within the single-cell proteomic landscape of human BM HSPCs. We hypothesized that by simultaneous quantification of cell surface and intracellular proteins we could infer the lymphoid lineage potentials and consolidate conflicting observations in human lymphopoiesis. In doing so, we are able to infer the lymphoid lineage potentials in poorly defined cellular compartments within the human hematopoietic hierarchy. This approach reveals TdT⁺ hematopoietic progenitors with lymphoid-primed proteomic and epigenetic landscapes within the canonical human GMP immunophenotypic compartment. Prospective isolation of this putative lymphoid-primed progenitor population via CD84ˡᵒ expression confirms its robust lymphoid potential in functional differentiation assays that is equivalent to that of human LMPPs isolated in parallel. Thus, our data demonstrate the utility of bottom-up interrogation of human systems to define a population based on multi-omic molecular profiling while identifying a significant source of human bone marrow lymphoid progenitors within a presumed myeloid-committed compartment.

## Results

### Single-cell proteomic map of human bone marrow HSPCs

To create a single-cell proteomic map of human BM HSPCs, we expanded our highly multiplexed CyTOF mass cytometry screen[21], enabling quantification of 351 surface protein molecules and 83 intracellular targets, including transcription factors (TFs), histone modification markers and metabolic enzymes[22] (Supplementary Data 1). While the total 434 targets were split across 15 different staining panels, all the panels included a conserved set of molecules (Supplementary Fig. 1A). Conserved molecules included seven cell surface proteins (CD10, CD34, CD38, CD45, CD45RA, CD90, CD123) used conventionally to define human HSPC types and 4 additional lineage-associated proteins—CD71, also known as transferrin receptor 1, for erythroid lineage[23–25], SATB1 for early lympho-myeloid progenitors[1,26–28], TdT for lymphoid lineage[1,11,23], and IRF8 for DC and monocyte lineages[29–31]. In total, we analyzed 556,226 CD34⁺ HSPCs from 3 different donors (Donor 1, 2, 3) and minimized technical batch effects via two rounds of split and pooling to barcode donor and panel information (Experimental Workflow in Supplementary Fig. 1B).

Using a stringent thresholding strategy to reduce the possibility of false positives, we identified 81 screen antigens that are detected in at least 0.1% of CD34⁺ HSPCs (Fig. 1B, Supplementary 2A). To infer the co-expression data of the 81 targets, grouped cells across the 15 different panels into micro-clusters using the conserved panel and then meta-clustered these micro-clusters using the median expression of both the conserved and screen antigens (Supplementary Fig. 2B, See Methods for details). As a result, we detected 10 clusters within the CD34⁺ HSPCs and annotated the clusters with prior knowledge, named here A1 to A10 (Fig. 1C, D, Supplementary 2C).

Consistent with previous RNA-sequencing (RNA-seq)[1,23] and the assay for transposase-accessible chromatin using sequencing (ATAC-seq)[2] studies of human HSPCs, we observed CD38lo progenitors to be tightly interconnected with each other, whereas more differentiated CD38hi progenitors were disconnected (Fig. 1D, Supplementary Fig. 2B). This observation suggests only subtle differences in the regulatory networks and gene expression of undifferentiated progenitors, consistent with a continuum of low-primed undifferentiated cells (CLOUD) in hematopoiesis[1]. Among CD38hi progenitors, the CD10⁺ A10 cluster (B committed progenitors) was embedded distantly from other TdT expressing clusters A8 (Intermediate lymphoid/myeloid progenitors) and A9 (Lymphoid progenitors) (Fig. 1C) in the dimensionally reduced Uniform Manifold Approximation and Projection (UMAP)[32] space generated with all markers in the panels (Fig. 1D, right, Supplementary Fig. 2D, Supplementary Data 4). This divergence may be due to rapid protein landscape changes as progenitors progress into B cell commitment. Thus, this observation corroborated our hypothesis for the existence of lymphoid progenitors between CD38lo LMPPs and CD10⁺ progenitors.

Protein-level cell-cycling and epigenetic states of the clusters also correspond to the degree of differentiation. As expected, the most

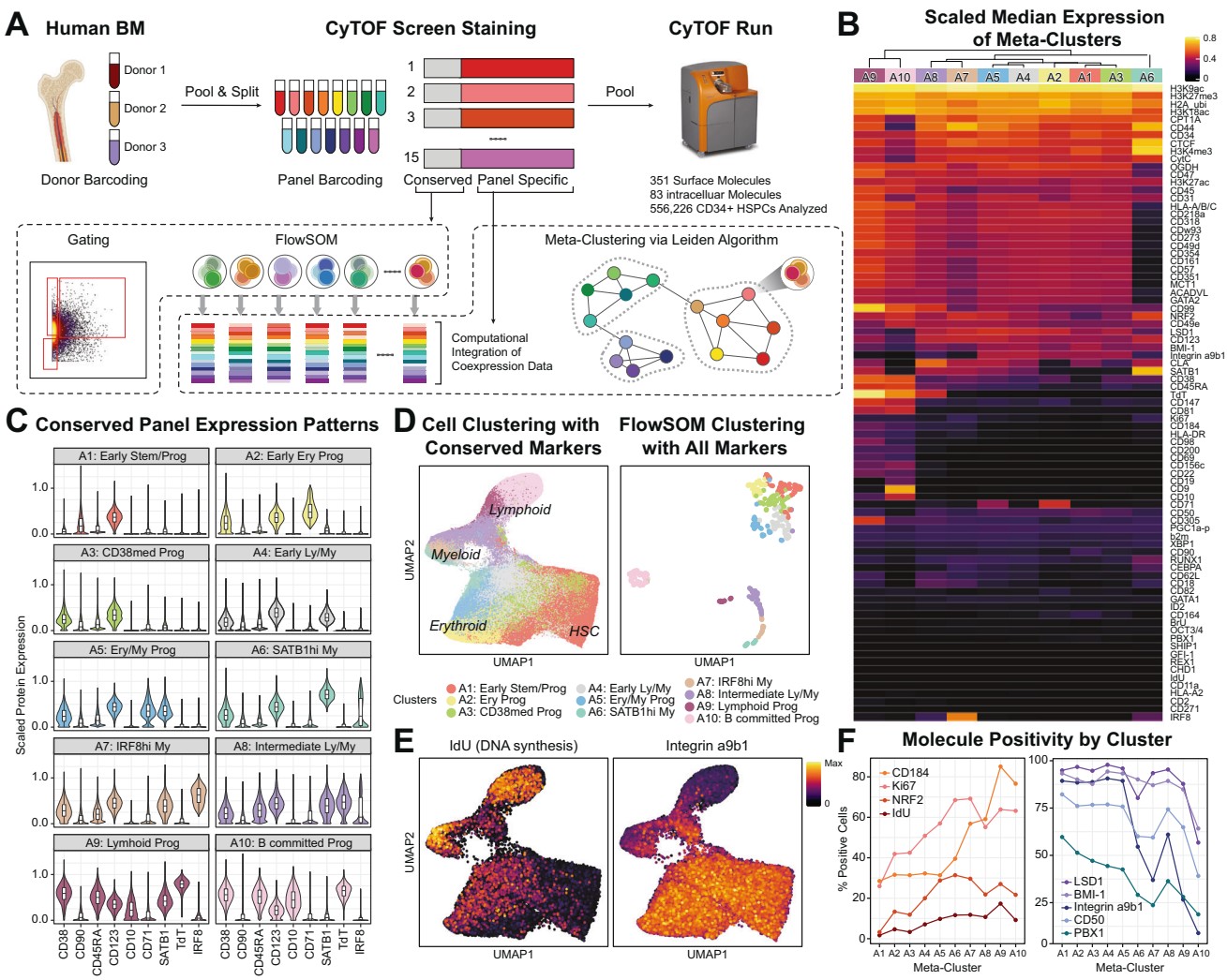

**Fig. 1 | Single-cell proteomic map of human bone marrow HSPCs.**
**A** Experimental overview of the single-cell proteomic screen. Total three healthy human bone marrow were analyzed for the proteomic screen. Created with BioRender.com, released under a Creative Commons Attribution-NonCommercial-NoDerivs 4.0 International license. **B** Heatmap of scaled median expression of molecules that were detected in at least 0.1% of CD34⁺ HSPC by meta-clusters. Columns and rows are hierarchically clustered. Meta-cluster annotation is in (**C**). **C** Violin plots of the conserved core panel protein expressions by meta-clusters. Expression levels are normalized to the 99.9th percentile of each molecule. Each box plot represents the median as the center line and the 1st and 3rd quartile as the top and bottom edge, with whiskers representing the maximum and minimum

values. Number of cells for each meta-cluster: A1 = 123,791, A2 = 49,324, A3 = 69,972, A4 = 84,206, A5 = 49,231, A6 = 15,403, A7 = 22,047, A8 = 42,362, A9 = 18,943, A10 = 80,497. Source data is available at Dryad https://doi.org/10.5061/dryad.xgxd254jp. **D** UMAP of all CD34⁺ cells (left) and of FlowSOM clusters (right). Cell level UMAP was created with the conserved panel markers measured in each cell and FlowSOM cluster-level UMAP was created with median expression levels of all molecules of each cluster. **E** UMAP of all CD34⁺ cells colored by IdU staining (left) or Integrin a9b1 staining (right). **F** Percentage of positive cells expressing protein molecules that are increasing (left) or decreasing (right) along the hematopoietic differentiation.

undifferentiated cluster A1 (Early stem/progenitors) had the lowest proliferation marker measurements, including both the lowest 5-Iodo-2'-deoxyuridine (IdU) incorporation (Fig. 1E, Supplementary 2E), which directly labels active DNA synthesis in S phase[33], and Ki67 (Supplementary Fig. 2F), whereas more differentiated clusters had higher proliferation (Fig. 1F, Supplementary 2F). H3K27ac histone modification, which marks active enhancers, exhibits a similar pattern (Supplementary Fig. 2F). In contrast, the polycomb complex protein BMI-1 was highest in A1 and decreased gradually in more differentiated clusters (Fig. 1F, Supplementary 2E), as previously reported[34,35].

Interestingly, similar trends appeared with several adhesion molecules. Integrin a9b1 heterodimer is an adhesion molecule suggested to play a role in HSPC cell adhesion to endosteal osteoblast[36], and exhibited decreased expression level along the differentiation (Fig. 1E, Supplementary 2D). While up to 90% of cells in the earlier clusters (A1 - A5) express Integrin a9b1, this fraction drops to ~ 50% of

cells in the intermediate A6, A7, and A8 clusters, and to less than 5% in the most differentiated A10 cluster (Fig. 1E, F, Supplementary 2D). Similar, if weaker, trends were observed for CD50 and CD164 adhesion molecules (Fig. 1F, Supplementary 2E), emphasizing the link between cell-cell interactions in the stem cell niche to the differentiation processes. Overall, this highly multiplexed single-cell proteomic screen provides deep cellular information on the cell states of human hematopoiesis, via integrating expression patterns of surface molecules, TFs, metabolic states, and epigenetic regulators. Furthermore, the protein molecules highlighted in this screen were used for subsequent analyzes and prospective isolation.

### Identification of the proteomic signatures of human BM HSPC populations

While we identified 81 substantially expressed targets from the original screen, we asked if we could identify a smaller subset of these molecules

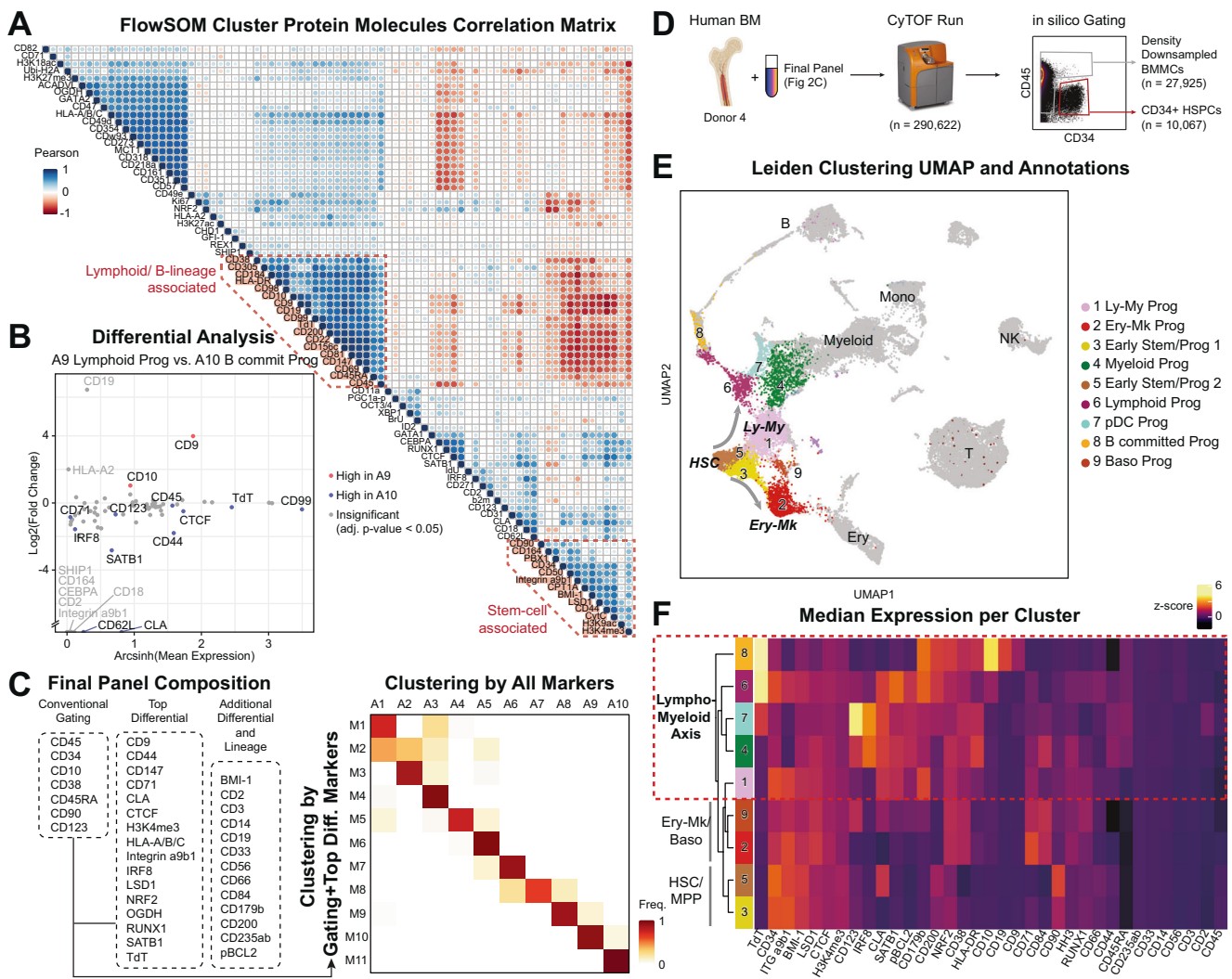

**Fig. 2 | Identification of the proteomic signatures of human BM HSPC populations. A** Pairwise correlation matrix of all positively detected protein molecules from the screen. Pearson correlation was measured at FlowSOM cluster-level. **B** Differential analysis of protein expression from CyTOF measurement between A9 Lymphoid progenitors and A10 B-committed progenitors Meta-clusters. Differences in the distribution of molecules were calculated in equally subsampled populations using the two-sample KS test and *p*-value was adjusted with Bonferroni correction. Source data for the figure is provided as a Source Data file. Results of all differential analysis are provided as Supplementary Data 2. **C** Protein targets in the final panel (left) and confusion matrix of comparing meta-clustering done with all

markers (A1 – A10) and minimal set of markers (M1 – M10) (right). **D** Workflow for the follow-up CyTOF experiment. One additional bone marrow (Donor 4) was analyzed to validate the screened and selected protein targets. A different healthy donor bone marrow was used and CD34⁻ cells were also included after density-downsampling. Created with BioRender.com, released under a Creative Commons Attribution-NonCommercial-NoDerivs 4.0 International license. **E** UMAP of BMMCs. Only CD34⁺ compartment is colored by their cluster and the rest CD34 compartment is colored gray. **F** Median expression heatmap of protein markers per cluster.

---

that captured the variance of this larger dataset. To look for redundancy, we calculated the Pearson correlations between all pairs of targets, detecting proteins with highly correlated expression patterns (Fig. 2A). Some examples of the most evident correlated proteins observed are lymphoid/B-lineage-associated, including TdT, CD10, CD19, CD9, CD22, and CD200, and stemness-associated proteins, including CD34, CD90, CD164, and PBX1, implying cell-state specific protein expression modules. Given this large amount of structure in the protein expression, we further reasoned that the protein landscape of HSPCs can be recapitulated with a reduced set of representative protein targets from each of these highly correlated groups. To this end, we nominated representative proteins with a differential analysis aimed at capturing the statistically significant differences among protein landscapes of meta-clusters (Fig. 2B, Supplementary Data 2). Putting all of this together, we selected the 23 of the most informative protein molecules from the screen that are either used for conventional HSPCs

selection (i.e., gating, Fig. 2C, Conventional Gating column) or were identified in our analysis of differentially expressed protein molecules with minimum redundancy (Fig. 2C, Top differential column). Using only these 23 protein molecules, we could recreate original clustering, which used all detected targets from the screen (Fig. 2C; right). We finalized our panel with additional lineage markers (Fig. 2C, Additional Differential and Lineage column) so as to complete the hematopoiesis trajectories with the more mature, CD34⁻ bone marrow mononuclear cells (BMMCs).

To assess this unified analysis panel, we quantified the protein expressions on a total 290,622 additional cells from another healthy human bone marrow (Donor 4) by mass cytometry. For the emphasis on the hematopoiesis, we enriched HSPCs *in silico* by gating CD34⁺ HSPCs and CD34⁻ BMMCs and downsampled the latter population (Workflow in Fig. 2D). A nearest-neighbor graph analysis with Leiden-clustering demonstrated that the reduced number of single-cell proteomic

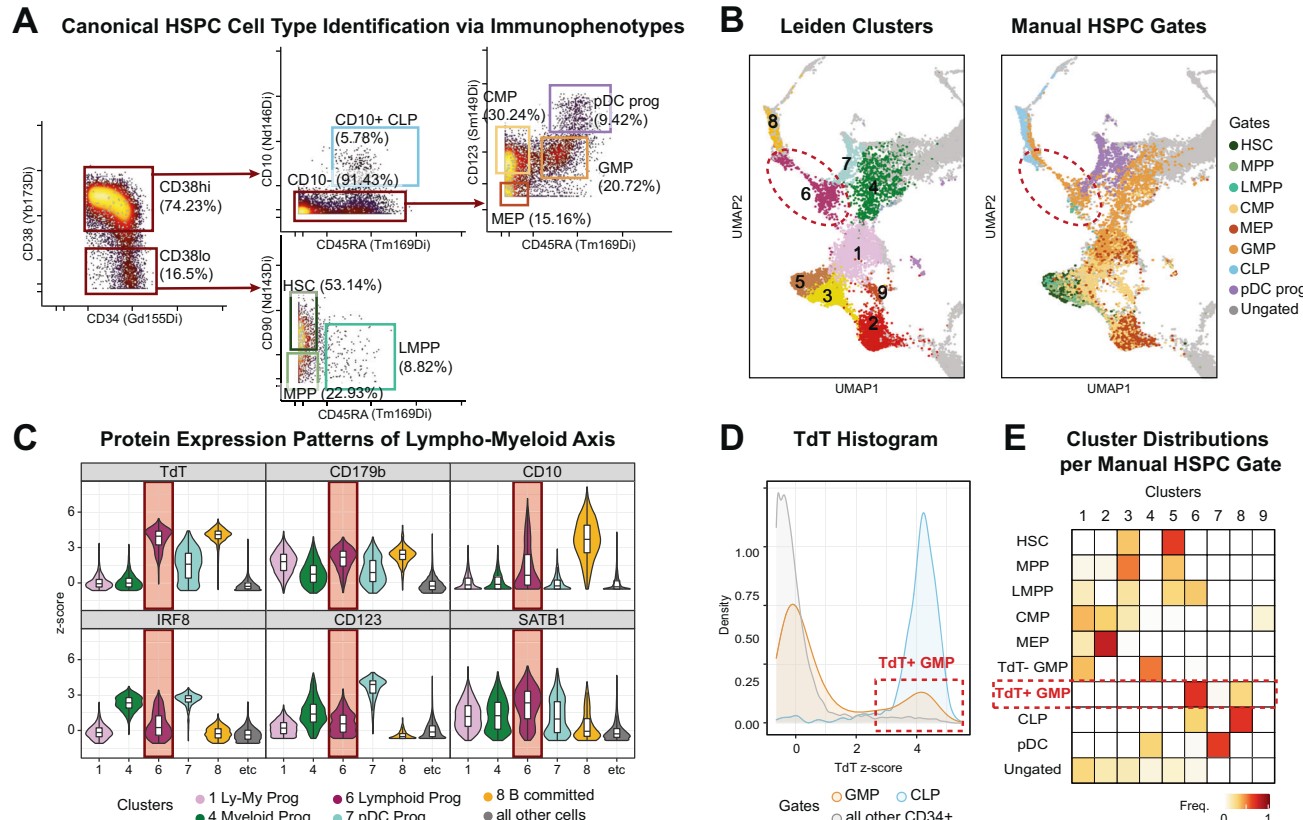

**Fig. 3 | Lymphoid proteomic features identified within the conventional granulocyte-monocyte progenitor compartment. A** Gating scheme of conventional HSPC cell types on CyTOF. Pregate: Singlet Live Lin- CD45$^{lo}$ CD34$^+$ cells ($n$ = 10,167) from Fig. 2D. **B** UMAP from Fig. 2E focused on CD34$^+$ compartment. Colored by Leiden clusters (left) or manually gated cell types from Fig. 3A (right). Cluster 6 Lymphoid Progenitor is circled in red dotted line. **C** Violin plots of lympho-myeloid lineage-associated protein expression patterns. Shown clusters are of lympho-myeloid lineages and all other cells are represented as gray. Each box plot represents the median as the center line and the 1st and 3rd quartile as the top and bottom edge, with whiskers representing the maximum and minimum values. Number of cells for each cluster: Cluster 1 = 2413, Cluster 4 = 1240, Cluster 6 = 756, Cluster 7 = 558, Cluster 8 = 491, all other cells (Clusters 2, 3, 5, 9) = 4546. Source data are provided as a Source Data file. **D** TdT protein expression histogram of GMPs, CLPs, and all other CD34$^+$ cells. Source data are provided as a Source Data file. **E** Confusion matrix comparing manually gated cell types to Leiden clusters. Each row is normalized.

features successfully captured both the heterogeneity of the proteomic landscape among CD34$^+$ HSPCs and the continuum into mature immune cells (Fig. 2E, Supplementary Fig. 3A). Within the CD34$^+$ compartment, we were able to annotate 9 clusters (Fig. 2E) with their protein expression patterns (Fig. 2F, Supplementary Fig. 3B, Supplementary Data 5). Of note, we observed bifurcation of Cluster 2 Erythro-Megakaryo (Ery-Mk) Progenitors and Cluster 1 Lympho-Myeloid (Ly-My) Progenitors immediately after exit from Clusters 3 and 5 Early Stem and Progenitors (Fig. 2E). Unlike the traditional HSPC model with CMP for erythro-megakaryo and myeloid lineages and CLP for lymphoid lineages, our analysis suggests a shared proteomic phenotype of lymphoid progenitors and myeloid progenitors which separate them from Ery-Mk progenitors. In fact, the Ly-My versus Ery-Mk bifurcation is also observed in transcriptomic and epigenetic landscapes from human HSPC single-cell RNA-seq[1,23] and ATAC-seq[3] studies, respectively. As the molecular phenotypes in three different modalities coincide, we speculate the functional differentiation potentials to reflect the phenotypic Ly-My versus Ery-Mk bifurcation. Furthermore, we presume the lymphoid progenitors to be sharing phenotypes that have historically been assigned to myeloid progenitors.

**Lymphoid proteomic features identified within the conventional granulocyte-monocyte progenitor compartment**

Given the differences between the traditional human hematopoiesis model and our data-driven clustering, we further examined the cluster assignments of canonical HSPC cell types. CD34$^+$ HSPC cell types were annotated based on conventional gating schemes (Fig. 3A). We note that our usage of the HSPC cell type names is to refer to the empirical populations with conventional surface immunophenotypes. The most striking observation was found in the granulocyte-monocyte progenitor (GMP, CD34$^+$CD38$^+$CD10$^-$CD45RA$^+$CD123$^{med}$) compartment, as a substantial subset of GMPs were annotated as Cluster 6 Lymphoid Progenitors (Fig. 3B). Cluster 6 was identified as such based on its high expression levels of lymphoid development-associated proteins, such as TdT and intracellular CD179b, also known as lambda 5 surrogate light chain[37] (Fig. 3C). Interestingly, while we detect intracellular CD179b (Gene name: IGLL1) protein expression in Cluster 1 Ly-My progenitors and Cluster 6 Lymphoid progenitors, before Cluster 8 B committed progenitors, the mouse homolog *Igll1* has been shown to be restricted in B committed progenitors[38–40]. This observation suggests possible differences in gene regulation between humans and mice in lymphoid development. In addition, Cluster 6 exhibited lower expression levels of myeloid-associated TF IRF8 and interleukin-3 receptor CD123 in comparison to more apparent myeloid progenitor Cluster 4 and pDC progenitor Cluster 7 (Fig. 3C). Moreover, trajectory-inference using PAGA algorithm[41] positioned Cluster 6 Lymphoid Progenitor between Cluster 1 Ly-My Progenitors and the Cluster 8 B-committed Progenitors (Supplementary Fig. 4), further corroborating our annotation.

To delineate the lymphoid progenitors among the cells displaying the conventional GMP surface immunophenotype, we focused on the characteristic expression of TdT of the Cluster 6. TdT, terminal

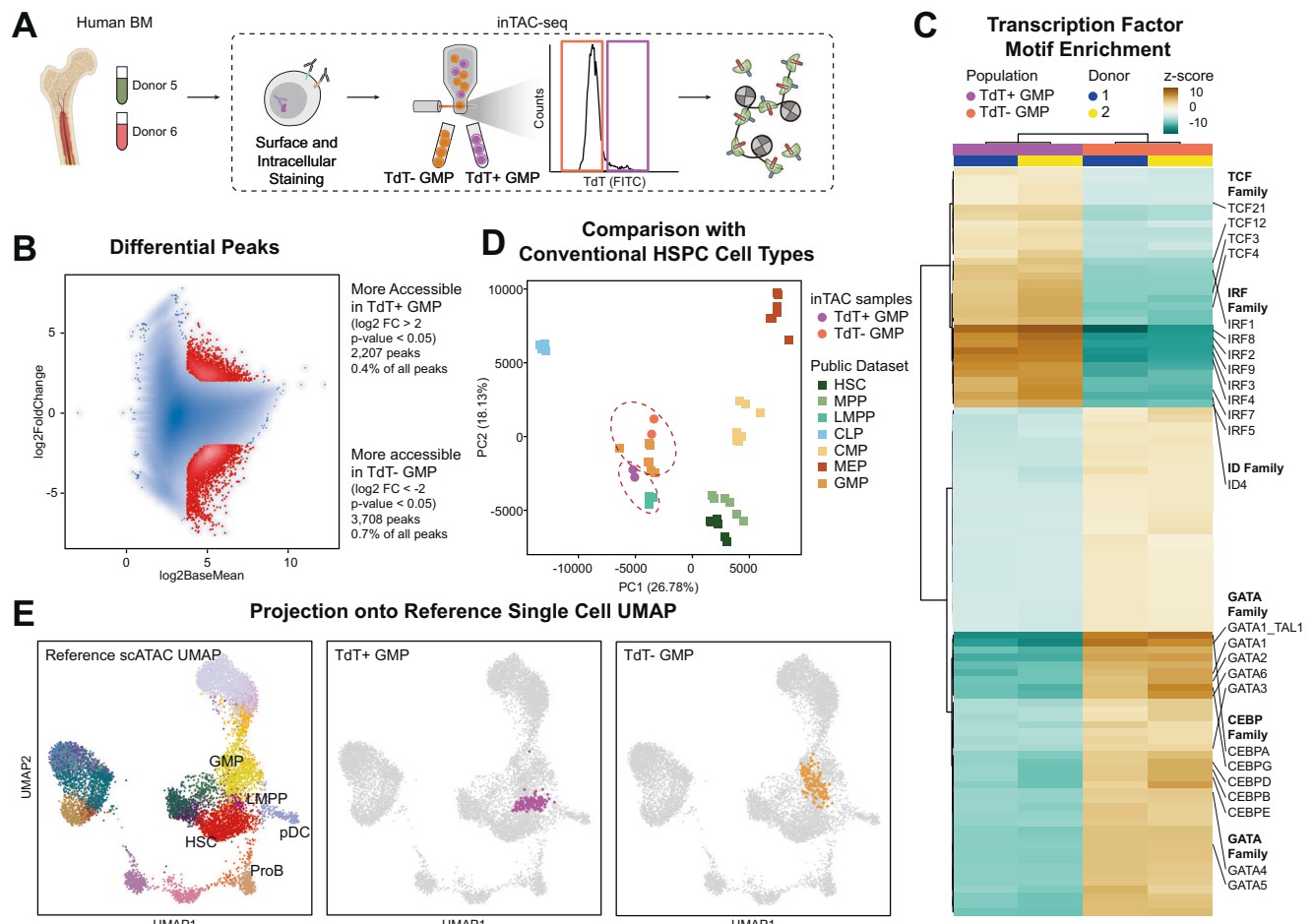

**Fig. 4 | TdT⁺ subset of human GMPs exhibit a lymphoid-primed chromatin accessibility landscape.** **A** Workflow of inTAC-seq. Two healthy human bone marrow (Donor 5 - 6) were sorted for TdT⁻ and TdT⁺ GMPs. GMPs were gated based on surface markers and then sorted into TdT⁺ or TdT⁻ population by TdT intra-cellular staining. Created with BioRender.com, released under a Creative Commons Attribution-NonCommercial-NoDerivs 4.0 International license. **B** Differential analysis of all peaks detected in TdT⁺ GMP and TdT⁻ GMPs. Differential analysis was performed via DESeq2 using the Wald test. *P*-values were first attained by the Wald test and were adjusted with Benjamin–Hochberg (BH) method for multiple comparisons. **C** Heatmap of ChromVAR transcription factor enrichment scores in differentially accessible peaks between TdT⁺ GMPs and TdT⁻ GMPs. **D** Principal component analysis of chromatin accessibility from inTAC-seq data from this study (circles) with publicly available sorted bulk HSPC ATAC-seq data (square). **E** UMAP projection of TdT⁺ GMP (middle) and TdT⁻ GMP (right) inTAC-seq data onto the reference BMMC scATAC-seq data (left). Cell types on the reference UMAP are annotations from the original data set.

deoxynucleotidyl transferase, is functionally intrinsic to VDJ recombination during lymphoid development and has been identified as the characteristic lymphoid gene in previous studies[1,11,23]. Furthermore, a recent study in mice demonstrated high TdT expression in lymphoid-biased MPP4s and CLPs but no expression in GMPs[42]. In contrast, the expression level of TdT in the conventional human GMP compartment was clearly bimodal (Fig. 3D) and TdT⁺ subset had the expression level as high as CLPs (Supplementary Fig. 3C). Subsequent classification of TdT⁺ subset of GMP surface immunophenotype cells (named here TdT⁺ GMPs) highly corresponded to Cluster 6 based on their proteomic profile (Fig. 3E, Supplementary 3D). Of note, TdT⁺ GMPs comprised Thus, despite its presumed myeloid identity based on the surface immunophenotype, TdT⁺ GMPs appeared to be lymphoid progenitors.

We further gathered additional BM CD34⁺ HSPC data that were collected on CyTOF from Favaro and Glass et al.[43]. and quantified the frequency of TdT⁺ GMPs among all CD34⁺ HSPCs. TdT⁺ GMPs comprised 3.20% (standard deviation 1.00%) of total CD34⁺ HSPCs. This frequency was significantly higher than the frequency of LMPPs (mean frequency: 0.28%, paired t-test *p*-value: 1.06×10⁻⁵) and CLPs (mean frequency: 1.82%, paired t-test *p*-value: 0.031) (Supplementary Fig. 3E).

## TdT⁺ subset of human GMPs exhibit a lymphoid-primed chromatin accessibility landscape

Considering the expressions of lymphoid-specific proteins such as TdT, we reasoned that TdT⁺ GMPs must have already been primed for lymphoid developmental programs. Thus, we assessed the lineage-specific chromatin accessibility of TdT⁺ GMPs to examine their developmental potentials. Since TdT is an intracellular protein, we utilized inTAC-seq[44] to purify target cells based on immunochemistry staining of TdT and assessed the transposase-accessible chromatin from two healthy bone marrow donors (Donor 5, 6). While they both subsets are considered GMPs by their surface immunophenotypes, TdT⁺ GMPs and TdT⁻ GMPs had 5,915 statistically significant differentially accessible regions between them (Fig. 4B). To delineate which transcription factors were associated with these differentially accessible chromatin regions, we applied ChromVAR[45] to calculate TF motif enrichment scores. TFs associated with lymphoid development, such as *TCF3*, *TCF4*, *IRF4*, *IRF8*, and *ID4*, were highly enriched in TdT⁺ GMPs (Fig. 4C), indicating lymphoid priming of this population. In contrast, multiple members of *GATA* and *CEBP* TF families, which are known as canonical myeloid lineage TFs, are strongly enriched in the open chromatin of TdT⁻ GMPs (Fig. 4C). Interestingly, we previously identified an enrichment for TCF4 motif accessibility in LMPPs biased towards CLP

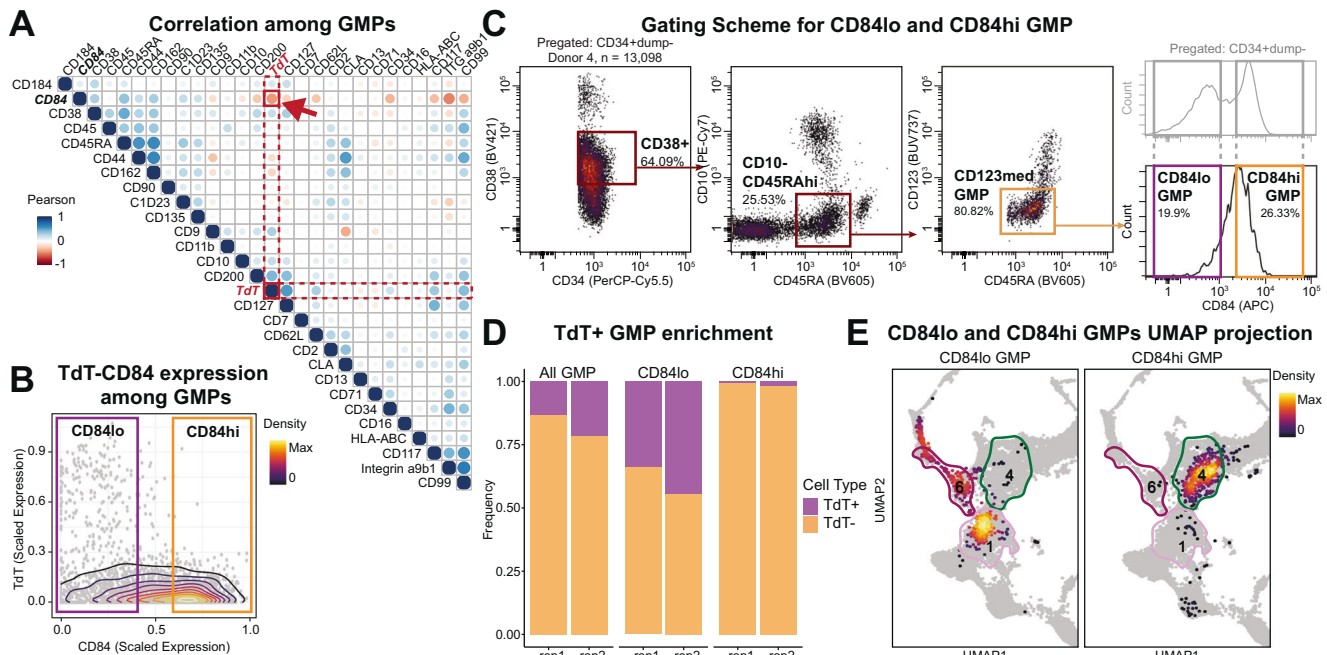

**Fig. 5 | Low CD84 surface expression is a surrogate surface phenotype for TdT+ lymphoid-primed progenitors. A** Pairwise correlation matrix of candidate surface protein molecules and TdT among GMPs. Pearson correlation was measured at cell level. **B** Biaxial plot of TdT and CD84 protein expressions in conventional GMPs. Boxes represent CD84 gates. **C** FACS gating scheme for CD84$^{lo}$ GMPs and CD84$^{hi}$ GMPs. **D** Frequency of TdT+ GMPs and TdT− GMPs in gated populations (Donor 4, two different experiments). Source data are provided as a Source Data file. **E** Projection of CD84$^{lo}$ GMPs and CD84$^{hi}$ GMPs in the proteomic UMAP from Fig. 2E. Outlines of Cluster 1 Early lympho-myeloid progenitors, Cluster 4 Myeloid progenitors, and Cluster 6 Lymphoid progenitors are drawn to represent the boundary of each cluster.

differentiation, and an enrichment for CEBPE motif accessibility in LMPPs biased towards GMP differentiation[2]. These same trends were recapitulated in the TDT+ GMPs and TDT− GMPs in our own data.

We then compared the chromatin landscape of TdT+/− GMPs to a bulk human HSPC ATAC-seq dataset[46] comprising canonical HSPC cell types identified by surface proteins. When we visualized our data with this data using Principal Component Analysis (PCA), TdT+ GMPs and TdT− GMPs straddled the canonical GMPs from the public dataset, with TdT+ GMPs located more proximal to LMPPs (Fig. 4D). This observed grouping and overall proximity of TdT+ GMPs to LMPPs and TdT− GMPs with canonical GMPs was also observed using UMAP visualization (Supplementary Fig. 5). We next compared the chromatin landscape of TdT+ GMPs from inTAC-seq data to a single-cell ATAC-seq data set that spanned the continuum of HSPC differentiation states, independent of the surface-based cell type identification. To achieve this, we simulated single-cell data from our bulk data by subsampling pseudo-single-cell data, then projected these simulated single cells into the original UMAP generated from the reference data (Fig. 4E). We observed that TdT+ GMPs overlap with clusters annotated as LMPPs in the direction towards pDCs and Pro-B cells. In contrast, TdT− GMPs overlap with the cluster annotated as GMPs in the reference dataset. Together, the chromatin accessibility of TdT+ and TdT− GMPs suggests that TdT+ GMPs are lymphoid-primed progenitors closer to LMPPs. The myeloid-specific progenitor identity presumed in the surface phenotypic GMPs seems to be restricted to TdT− GMPs.

**Low CD84 surface expression is a surrogate surface phenotype for TdT+ lymphoid-primed progenitors**

While TdT is an enzyme with a well-known function in lymphoid development, the measurement of this intracellular protein requires fixation and permeabilization to be detected in primary human cells. Therefore, to prospectively isolate live TdT+ GMPs, we stained an additional aliquot of BMMCs (Donor 4) with a mass cytometry panel focusing on cell surface proteins identified as candidates from our

previous mass cytometry experiments or suggested from literatures as lymphoid markers. Then, we computed Pearson correlation coefficients between cell surface proteins and TdT expression levels in the conventional human GMP compartment. (Fig. 5A). Among candidates, CD84, also known as signaling lymphocyte activation molecule (SLAM) family member 5[47], had one of the highest absolute levels of correlation ($R = -0.429$) and a clear bimodal expression pattern in HSPCs (Fig. 5B, C). Expression of CD84 was previously associated with lineage commitment among CD34+ progenitors[48]. Other candidates with comparable correlations captured only a small fraction of the TdT+ lymphoid primed population and had a less stark distinction between positive and negative populations (Supplementary Fig. 6A, B). Of note, we have analyzed inTAC-seq data with reference scATAC-seq data to identify TdT+ GMP marker genes (with threshold of FDR < 0.05 and log2 fold change > 1), but none of the identified marker genes was cell surface protein. Gating on CD84 enriches TdT+ GMPs by 2.3-folds compared to in all GMPs, resulting in 39.2% of CD84$^{lo}$ GMPs expressing TdT (Fig. 5D). While our CD84$^{lo}$ GMP gate selected 73.2% of all TdT+ cells among GMPs (Supplementary Fig. 6A), we note that the imperfect selection of TdT+ cells was due to the stringent gating scheme (Fig. 5B) and there were less than 2% of TdT+ GMPs in CD84$^{hi}$ GMP gate (Fig. 5D). We confirmed that the protein expression patterns of TdT and CD84 were conserved in other donors by additionally analyzing 8 bone marrow samples in a different dataset (Supplementary Fig. 6D)[43]. Furthermore, CD84$^{lo}$ GMPs and CD84$^{hi}$ GMPs exhibited a clear distinction in their cluster assignments, with CD84$^{lo}$ GMPs corresponding primarily to Cluster 6 lymphoid progenitors and Cluster 1 lympho-myeloid progenitors (Fig. 5E). As Cluster 1 lympho-myeloid progenitors preceded Cluster 6 in the trajectory analysis (Fig. 3D) and are mostly TdT-negative, we could infer that TdT−CD84$^{lo}$ GMPs were likely earlier progenitors with lymphoid potentials before TdT upregulation. In contrast, CD84$^{hi}$ GMPs were almost exclusively assigned to Cluster 4 myeloid progenitors (Fig. 5E). Given these results, we concluded that CD84$^{lo}$ GMPs could represent the lymphoid-primed progenitors in the

human lympho-myeloid axis that could be isolated for downstream cellular assays.

### CD84[lo] GMPs yield robust multi-lymphoid output with in vitro differentiation assays

To functionally validate the lymphoid developmental potentials presumed from the multi-omic molecular characterization of CD84[lo] GMPs, we conducted in vitro differentiation assays with the OP9-DL4 co-culture system, which can support both T and NK lineage differentiation from human HSPCs[49,50]. We prospectively isolated CD84[lo] GMPs from two healthy donors bone marrows (Donor 7, 8) along with CD84[hi] GMPs, LMPPs, and CLPs (Supplementary Fig. 7A). Consistent with our hypothesis, CD84[lo] GMPs yielded robust T and NK lineage output (Fig. 6B, C). By week 3, CD84[lo] GMPs and LMPPs proliferated significantly more than CD84[hi] GMPs or CLPs (Fig. 6B). While CD84[lo] GMPs, LMPPs and CLPs all gave rise to lymphoid progeny, CD84[hi] GMPs only differentiated into CD14[+] or CD15[+] myeloid cells (Fig. 6B). By week 5, CD84[lo] GMPs and LMPPs continued to expand, but CD84[hi] GMPs and CLPs yielded few progeny cells (Fig. 6C, S7B). CD84[lo] GMPs and LMPPs had the similar potential for the generation of T lineage cells expressing thymocyte markers, CD1a and CD7 (Fig. 6C), and T cell markers, CD4 and CD8 (Fig. 6C). To confirm the all lymphoid potentials, we used OP9 co-culture system to measure B cell differentiation potentials[51]. CD84lo cells robustly proliferated in OP9 co-culture media in the presence of IL-7 and developed into CD19[+] B cells. On the other hand, CD84[hi] cells could not yield any B cells. At the same time, CD10[+] CLPs were less proliferative but more effectively differentiated into CD19[+] B cells (Fig. 6D). We note that CD38[lo] LMPPs did not yield CD19[+] B cells in our culture condition with IL-7 only. It has been reported earlier progenitors require FLT3 ligand (FLT3L), while more downstream lymphoid progenitors which are already responsive to IL-7 do not require FLT3L[52]. Thus, we conclude that while CD38[lo] LMPPs are the earliest lymphoid progenitors that are not yet responsive to IL-7, CD84[lo] cells have robust lymphoid potential to all lymphoid lineages (T, B, NK) in vitro, and CD10[+] CLPs are highly efficient B cell progenitors. On the other hand, CD84[hi] myeloid progenitors are devoid of any lymphoid potentials.

To further quantify the clonal nature and the potency of T lineage potential in these cells, we conducted a limiting dilution assay in the OP9-DL4 system and fitted a generalized linear model GLM using ELDA software[53]. The estimated frequency (f) of T cell progenitor in CD84[lo] GMP population was 0.194, similar to that of LMPP population (f = 0.263, pairwise test p-value = 0.345), but significantly higher than CD84[hi] GMP (f = 8.87e-4, p-value = 1.54e-25) and conventional CD10[+] CLPs (f = 1.11e-2, p-value = 1.09e-13) (Fig. 6G). Overall, CD84[lo] GMPs exhibited strong T lineage potential comparable to that of LMPPs, while CD84[hi] GMPs almost completely lack lymphoid potentials.

In addition, we assessed the erythro-myeloid potentials of CD84[lo] GMPs via colony formation assay with methylcellulose medium that supports erythroid (E), monocytic (M), and granulocytic (G) lineages differentiation. Consistent with their embedding within the single-cell proteomic landscape (Figs. 5E, 3B), CD84[lo] GMPs consisted of colony-forming unit-granulocytes and macrophages (CFU-GM) cells exclusively, generating approximately 1 GM colony for every 3 input cells (Fig. 6H, Supplementary Fig. 7C). This concurrent myeloid potential in CD84[lo] GMPs reconciles with the conventional GMP identity previously associated with these cells. LMPPs or CD84[hi] GMPs exhibited GM potential to a lesser amount and yielded a few erythroid colonies, suggesting some burst forming unit-erythroid (BFU-E) cells in the input (Fig. 6H, Supplementary Fig. 7D). The appearance of erythroid colonies also highlights the promiscuity in lineage priming across phenotypically similar progenitor populations. In contrast, the purity of GM colonies from the CD84[lo] GMPs highlights the near-optimal purification of lympho-myeloid progenitors by extensive multi-omic molecular characterization. Lastly, to assess whether the CD84[lo] cells are

lympho-myeloid bipotential progenitors or mixture of unilineage lymphoid progenitors and myeloid progenitors, we conducted a single-cell cloning assay of CD84[lo] cells with 3 bone marrow samples. We note that SGF15/2 culture[6] for multi-lineage readout was not successful with adult bone marrow HSPCs in our hands. Instead, we optimized OP9-DL4 co-culture system to measure CD7[+] lymphoid cells and CD14[+] or CD15[+] myeloid cells. Out of 312 wells cloned, 18.6% (58 wells/312 wells) were successfully cloned. Of the wells positively cloned, 43.4% (standard deviation = 10.0%) had both lymphoid and myeloid progeny, suggesting a substantial lympho-myeloid bipotentiality at clonal level among CD84[lo] cells. While the lymphoid-only or myeloid-only frequency showed a large donor-to-donor variation, the bipotentiality was observed consistently across three replicates (Fig. 6G). Compared to a previous study using human cord blood samples[6], this level of bipotentiality is only seen among LMPPs, corroborating our annotation of these cells as early lympho-myeloid progenitors in proteomic space. Altogether, our functional differentiation results validate the lymphoid potentials of CD84[lo] GMPs and demonstrate the prolonged coexistence of lymphoid and myeloid potentials in the human lympho-myeloid axis.

## Discussion

We present a comprehensive summary of the human BM HSPC proteome by quantifying both cell surface proteins and intracellular proteins, encompassing the canonical markers used for cell type identification as well as functional protein molecules that represent the functional state of the cell. Our screen identified 81 protein targets expressed in BM HSPCs and provides a basis on which canonical and modified definitions of HSPC populations can be compared. Using high dimensional proteomic data, we determined the cell types of the clustering results based on the functional proteins. This approach led to the discovery of TdT[+] lymphoid progenitors with the traditional immunophenotype of GMPs. Although previously regarded as myeloid committed within the canonical HSPC classification, TdT[+] GMPs exhibit lymphoid bias in both proteomic and epigenetic landscapes that are distinct from the rest of TdT[-] GMPs. Furthermore, we identified CD84[lo] as the surface phenotype to enrich TdT[+] GMPs for live-cell sorting, where CD84[lo] GMPs yield robust lymphoid output in cellular differentiation assays. In contrast, CD84[hi] GMPs were nearly devoid of TdT[+] GMPs and showed lack of lymphoid potentials. Thus, we report strong molecular lymphoid bias in TdT[+] GMPs and robust lymphoid potentials in CD84[lo] GMPs in the human bone marrow.

Our analysis also revealed the erythroid versus lympho-myeloid bifurcation in the proteomic landscape consistent with structures seen in single-cell transcriptomic[1,23] and epigenetic landscapes[3]. Unlike the traditional model with pan-myeloid (including erythroid) versus lymphoid bifurcation in early progenitors, we observe Ery-Mk progenitors (Cluster 2) and Ly-My progenitors (Cluster 1) immediately downstream of the CD38[lo] early stem/progenitors with multipotency (Cluster 3 and 5; corresponding to HSCs and MPPs in conventional HSPC cell types) (Fig. 2E). When we compared conventional gating schemes to the unsupervised clustering of HSPCs (Fig. 3E, Supplementary Fig. 3D), we observed cell types in the lympho-myeloid axis, such as CMPs, LMPPs, and GMPs, were dispersed over multiple clusters. This discrepancy between high-dimensional profiling and low-plex cell surface protein-based immunophenotyping in the lympho-myeloid axis emphasizes our current lack of understanding, and possibly over-generalizations, in current models of human early lympho-myeloid development.

In this study, we focused on the protein-level expression of lymphoid signatures, which led to the identification of TdT[+] GMPs that correspond to Cluster 6 (Figs. 3C, D). The frequency of TdT[+] GMPs among all CD34[+] HSPCs was 3.20%, which was 11.43 times higher than the frequency of LMPPs and 1.76 times more than of CLPs (Supplementary Fig. 3E). Thus, we have effectively doubled the source of lymphoid progenitors that can be identified in human bone

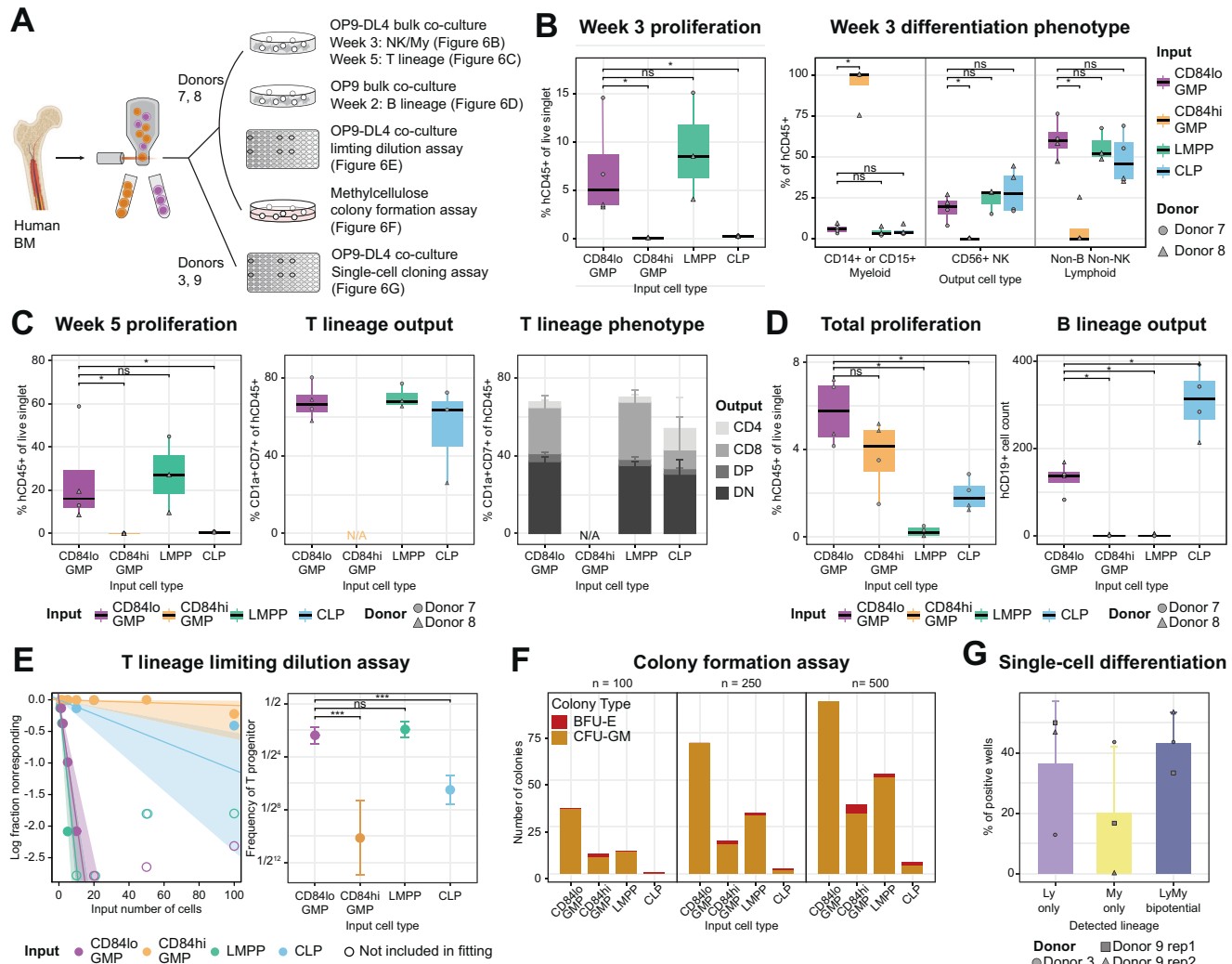

**Fig. 6 | CD84^lo GMPs yield robust multi-lymphoid output with in vitro differentiation assays. A** Workflow of in vitro differentiation assays. Bone marrow from Donor 7, 8 with two technical replicates per donor were used for bulk co-culture assays, limiting dilution assay, and methylcellulose assay and one sample of Donor 3 and two replicates from Donor 9 were used for single-cell cloning assay. Created with BioRender.com, released under a Creative Commons Attribution-NonCommercial-NoDerivs 4.0 International license. **B–D** Results of OP9/OP9-DL4 bulk co-culture. HSPCs from two donors, each with two technical replicates were used. Each box plot represents the median as the centre line and the 1st and 3rd quartile as the top and bottom edge, with whiskers representing the standard error. Standard error was not calculated for experimental group with two or less data points. Two-tailed Wilcoxon test was used and * represents *p*-value < 0.05. Source data are provided as a Source Data file. **B** Results of NK/My differentiation from OP9-DL4 bulk co-culture in week 3. Overall proliferation (left) and differentiation phenotypes (right). Exact *p*-values; for proliferation: CD84^lo GMP vs. CD84^hi GMP = 0.0286, CD84^lo GMP vs. LMPP = 0.4, CD84^lo GMP vs. CLP = 0.0286, for myeloid output: CD84^lo GMP vs. CD84^hi GMP = 0.0265, CD84^lo GMP vs. LMPP = 0.4, CD84^lo GMP vs. CLP = 0.486, for NK output: CD84^lo GMP vs. CD84^hi GMP = 0.0211, CD84^lo GMP vs. LMPP = 0.4, CD84^lo GMP vs. CLP = 0.486, for non-B non-NK lymphoid output: CD84^lo GMP vs. CD84^hi GMP = 0.0265, CD84^lo GMP vs. LMPP = 0.857, CD84^lo GMP vs. CLP = 0.343. **C** Results of T cell differentiation from OP9-DL4 bulk co-culture in week 5. Overall proliferation (left), percentage of T lineage output (middle), and T lineage subtype phenotypes (right). Exact p-values: CD84^lo GMP vs. CD84^hi GMP = 0.0294, CD84^lo GMP vs. LMPP = 0.857, CD84^lo GMP vs. CLP = 0.0286. Each section of stacked bars represents mean percentage of each output

population and error bars represent the standard deviation. **D** Results of B cell differentiation from OP9 bulk co-culture in week 2. Overall proliferation (left) and B lineage output (right). Exact *p*-values; for proliferation: CD84^lo GMP vs. CD84^hi GMP = 0.343, CD84^lo GMP vs. LMPP = 0., CD84^lo GMP vs. CLP = 0.0286, for B output: CD84^lo GMP vs. CD84^hi GMP = 0.0211, CD84^lo GMP vs. LMPP = 0.0211, CD84^lo GMP vs. CLP = 0.0286. **E** Ratio of T lineage progenitor from OP9-DL4 co-culture limiting dilution assay as a line graph of non-responding wells (left) and a summary dot plot (right). (Left) Each dot represents the observation of responding wells per each test condition. Empty dots represent wells data points no non-responding wells meaning all wells generated T lineage cells. The slopes of the lines are calculated as 1/(estimated progenitor frequency) and the shades represent the confidence intervals. (Right) Each dot represents the estimated progenitor frequency from limiting dilution assay in left. Error bars represent the confidence interval from the analysis. *p*-Values were calculated from pairwise likelihood ratio test. *** represents *p*-value < 0.005. *p*-Value for CD84^lo GMP vs. CD84^hi GMP = 1.54 × 10^{-25}, CD84^lo GMP vs. LMPP = 0.345, CD84^lo GMP vs. CLP = 1.09 × 10^{-13}. Source data are provided as a Source Data file. **F** Bar plots of methylcellulose colony-forming assay with input numbers of 100 (left), 250 (middle) and 500 (right) cells from each population. Source data are provided as a Source Data file. **G** Bar plot of the single-cell cloning assay results. A single CD84^lo GMP from 3 different bone marrow samples were sorted on OP9-DL4 mono-layer was cultured for 2 weeks and then measured for the existence of lymphoid (CD7^+) cells or myeloid (CD14^+ or CD15^+) cells. Error bars represent the standard deviation. Each symbol of the dot represents a replicate in the experiment. Source data are provided as a Source Data file.

marrow, expanding the sample availability for future lymphoid-targeted applications.

We noted that despite the distinct lymphoid-primed proteomic and epigenetic landscape, we were unable to define cell surface proteins that separate TdT[+] putative lymphoid progenitors exclusively. Instead, we isolated a broader population of CD84[lo] GMPs. Compared to the previously proposed lymphoid populations, our CD84[lo] GMPs appeared to be a superset of CD62L[hi] cells[12] and a subset of CD38[med] GMPs[6] (Supplementary Fig. 6C). While the exact differences among these generally overlapping populations should be investigated in future studies, we concluded that the CD84 is the most optimal surrogate for the following reasons. First, the CD84[lo] GMPs gate most efficiently enriched for TdT[+] GMPs (Supplementary Fig. 6A). Second, CD84 expression levels in BM HSPCs exhibited clear bimodal distribution by flow cytometry (Fig. 5C). And lastly, TdT[-] cells among CD84[lo] GMPs corresponded to the Cluster 1 Lympho-myeloid Progenitors, while CD84[hi] GMPs were identified as myeloid-specific Cluster 4 Myeloid Progenitors (Fig. 5E). Thus, selecting for CD84[lo] GMPs, successfully separates progenitors with lymphoid potential from the myeloid-specific progenitors. Still, it is possible that better prospective surrogates for these TdT[+] lymphoid progenitors could be identified in the future. In doing so, we may be able to better assess the level of lympho-myeloid multipotency along the human lympho-myeloid axis.

With in vitro differentiation assays, we have shown all lymphoid lineage potentials – T, B, and NK – of the CD84[lo] population. Especially for T cell potential, CD84[lo] population exhibited ~17 times higher frequency of T cell progenitor than that of CD10[+] CLPs (Fig. 6E), while CD10[+] CLPs exhibit more efficient B cell differentiation capacity (Fig. 6D). Thus, CD84[lo] cells and CD10[+] cells in humans strongly resembles the identification of Ly6D[-] all lymphoid progenitors (ALPs) in mice to distinguish from Ly6D[+] B cell-biased progenitor (BLP) in mice[20]. At the same time, the myeloid potential of CD84[lo] population could be compared to the human LMPPs or murine MPP4s[54]. Based on the high dimensional proteomic and epigenomic analyzes, we might postulate that TdT[+]CD84[lo] and TdT[-] CD84[lo] cells each resemble murine ALPs and MPP4s.

Furthermore, the single-cell cloning assay of CD84[lo] cells showed a substantial lympho-myeloid bipotentiality among CD84[lo] cells in vitro (Fig. 6G). This is surprising as Notta et al. demonstrated a significant reduction of multi-lineage progenitors in BM compared to the CB or fetal liver (FL) among Ery, Mk, My lineages[5]. The prevalence of Ly-My bipotential progenitors in adult bone marrow might imply that lymphoid and myeloid lineages are much more closely associated than we have imagined, even more so than the Ery-Mk-My lineages that have been traditionally grouped together. Considering that the functional roles of lymphoid and myeloid lineages are both for immune responses, compared to Ery lineage for circulation and Mk lineage for clotting and wound healing, it is reasonable that their evolutionary emergence and differentiation regulation are tied closely together.

We recognize that our in vitro functional differentiation assays may not precisely represent the in vivo potentials of the progenitors. Despite the utility of the xenograft models to study human hematopoiesis, models that support human T lineage development are rare, due to the complicated T cell development process in the thymus. The few existing models, withal, involve complex co-transplantation of human fetal thymic tissues, making them largely inaccessible[55]. Meanwhile, the in vitro differentiation assays we conducted might be more relevant to the clinical application settings. For example, recent studies have demonstrated ex vivo differentiation of human T lymphoid progenitors (HTLPs) from BM HSPCs for T cell reconstitution after bone marrow transplantation[56] and for T-cell-based immunotherapy[57]. Thus, the robust in vitro T cell differentiation from our lymphoid progenitors in this paper suggests that these cells would likely serve as an effective input for clinical applications. Nonetheless, we anticipate prospective studies combining lineage tracing and advances in human HSPC differentiation assays to determine the lineage potentials of progenitors in vitro and in vivo.

While our analysis focused on identifying lymphoid progenitors, we do not provide evidence supporting strict progenitor-successor relationships among canonical and newly discovered lymphoid progenitors. We trace this challenge to the heterogeneity in canonical HSPC cell types CMPs and LMPPs—the presumed predecessors of GMPs. The heterogeneities among CMPs have also been illustrated both in mice and humans[5,58,59]. As we append the lymphoid arm onto this framework, we observe both CMPs and LMPPs dispersed across several of these data-derived clusters (Fig. 3E). The top 2 most enriched clusters among conventional CMPs are Cluster 1 lympho-myeloid progenitors and Cluster 2 erythro-megakaryo progenitors, implying the myeloid (Mono-Gran) versus erythroid (Ery-Mk) bifurcation in CMPs. The myeloid progenitors among CMPs are clustered with other lymphoid progenitors (LMPPs, CD84[lo] GMPs) in Cluster 1 (Fig. 3C, Supplementary 2D), highlighting the similar phenotypes of myeloid and lymphoid progenitors among human HSPCs. Similarly, LMPPs, the other presumed progenitor of GMPs, are scattered across Clusters 1, 3, 5, and 6 spanning the entire lymphoid trajectory prior to CLPs. The dispersed appearance of LMPPs suggests that this surface phenotype marks a molecularly diverse set of cells. Hence, we determined that the canonical immunophenotypes of CMP or LMPP are inappropriate to define predecessor cell types to model progenitor-successor relationships in the lympho-myeloid axis.

Considering the significance of lymphoid development in healthy and malignant hematopoiesis, such as leukemia, bone marrow transplantation, and immune aging, we imagine our characterization of the human bone marrow lympho-myeloid axis to be the basis for future studies and applications. We expect future investigations to query the molecular mechanisms of fate decisions, which will provide the cues to intervene in malignant lymphopoiesis or to boost healthy lymphopoiesis. Beyond the insights to human lymphoid cell potential and cell identity, this study also provides a framework to quantify functional, lineage-associated proteins as surrogates for cell identity within the context of multi-modal single-cell molecular phenotyping. These single-cell molecular archetypes can then be associated with corresponding immunophenotypes for live-cell prospective isolation for functional interrogation and routine enumeration. Via this framework, we successfully reassessed the human lympho-myeloid axis and identified multi-lineage lymphoid progenitors in bone marrow. Moreover, we anticipate our approach of using protein-level molecular regulators of cell function as surrogates for lineage reporters to be expanded to various human tissues beyond the hematopoietic system.

## Methods
### Ethics statement
Our study complies with all relevant ethical regulations. Human bone marrow (BM) samples ($n = 9$) in this study were purchased and obtained as deidentified samples from AllCells (Alameda, CA) and StemExpress (Folsom, CA). Samples were collected by qualified clinicians from the posterior iliac crest of healthy and consenting donors following the vendors' IRB-approved protocols. Experimental donor details are included in Supplementary Table 1.

### Ex vivo labeling human bone marrow for CyTOF screen
Fresh BM aspirates were labeled for their biosynthesis[33]. Briefly, Fresh BM aspirates were immediately transferred to a 37 °C, 5% $CO_2$ incubator in T75 flask for 30 min prior to SOM3B labeling. A mixture of all three label molecules was added together and mixed thoroughly (final concentration; IdU (Sigma I7125) 100 μM, BRU (Sigma 850187) 2 mM, puromycin (P212121 58-58-2) 10 μg/mL), and then added to pre-warmed

bone marrow. Labeling was conducted in a 37 °C, 5% $CO_2$ incubator for 30 min before further processing of BM.

## Human bone marrow processing
Mononuclear cells were isolated from same day BM aspirates by using Ficoll-Paque plus density gradient media (Cytiva17-1440-03) to remove granulocytes and erythrocytes per manufacturer instructions. Bone marrow mononuclear cells (BMMCs) were utilized freshly isolated for the CyTOF proteomics screen and inTAC-seq or frozen in freezing medium (FBS (Omega Scientific FB-01) using 10% DMSO (Sigma-Aldrich D2650-100ML)) for use in further CyTOF, sorting, and functional assays. Cryopreserved BMMCs were thawed using thawing media (complete RPMI medium [RPMI 1640 (Gibco 21870092) supplemented with 10% FBS, Glutamax (Gibco 35050061) and 100 units/ mL of Penicillin and 100 μg/mL Streptomycin (Penicillin-Streptomycin, Gibco 15140122)], with 20 U/mL sodium heparin (Sigma-Aldrich H3149-100KU) and 0.025 U/mL benzonase (Sigma-Aldrich E1014-25KU).

## CD34$^+$ magnetic enrichment
CD34 MicroBead kit (Miltenyi 130-100-453) was used as manufacturer's instructions to enrich the CD34 compartment from BMMCs in the CyTOF screen and inTAC-seq. CD34- cells in the flow-through from wash steps in the protocol were also washed, counted for cell numbers to be added as a spike-in for the screen or frozen in freezing medium.

## CyTOF antibody preparations
Most CyTOF antibodies were acquired from our previous study[21]. Additional antibodies needed were conjugated using the MaxPar X8 Antibody Labeling kit per manufacturer instruction (Fluidigm 201300) or purchased from Fluidigm. Post-conjugation, each antibody was quality checked on positive and negative control cell lines or human PBMCs and titrated to an optimal staining concentration for $3 \times 10^6$ cells per test.

Panel 1 - 12 extracellular screening panels were prepared beforehand and lyophilized for storage. Each panel was prepared as a single-test master mix in total 100 μL with 100 mM D-(+)-Trehalose dihydrate (Sigma-Aldrich T9531) and 0.1X cell staining medium (CSM: PBS with 0.5% BSA (Fisher BP1600100) and 0.02% sodium azide (Sigma-Aldrich S2002-25G)) in double-distilled $H_2O$ (ddH2O). Prepared single-test aliquots were lyophilized in a vacuum chamber. Lyophilized panels were stored in −20 °C until usage. Before staining, each panel was reconstituted in 40 μL CSM, pipetted thoroughly, and filtered with Durapore 0.1 μm PVDF membrane filter (Millipore VVLP04700) for 2 min at 100 g to remove any possible precipitates in the antibody mix.

Core panel and Panel 13 - 15 intracellular screening panels were prepared on the day of experiment. For all panels, the master mix was filtered with Durapore 0.1 μm PVDF membrane filter for 2 min at 100 g before staining.

## CyTOF screen staining and acquisition (Workflow in Supplementary Fig. 1B)
For CyTOF screen, fresh mononuclear cells collected from three donors, each 50 ml of BM, were used for experiments.

For donor barcoding and viability staining, CD34$^+$ cells from each donor was supplemented with CD34$^-$ cells from the same donor upto total 15 x 1e6 cells in total 100 μL with cold FACS benzonase buffer (FBB, FACS buffer [PBS supplemented with 5% FBS and 20 U/mL sodium heparin] with 0.025 U/mL benzonase) in an 5 mL FACS tube. Cells were live-cell barcoded per donor[60]. Briefly, each donor sample was labeled with beta-2-microglobulin and CD298 antibodies conjugated with one of In113, Pt195, Pt196 isotopes for 30 min at room temperature. Monoisotopic (Pt194) cisplatin (1 μM) (Fluidigm 201194) was added for the final 5 min to stain non-viable cells[61]. Cells were washed with FBB and centrifuged for 5 min at 300 g, 4 °C. After removing supernatant by aspiration, cell pellets were resuspended

with residual volume and pooled into a single FACS tube and supplemented with CSM upto a total of 300 μL.

For Core panel extracellular staining, surface staining portion of the core panel was prepared as a 15-tests master mix in total 200 μL with CSM. Core panel was added to the pooled sample and stained for 30 min. Subsequently, 450 μL of FACS Buffer was added to quench staining and the sample was split into 15 FACS tubes with 60 μL of sample each, which represent the 15 screening panels.

For surface panel staining, each panel was prepared by reconstituting a lyophilized single-test mastermix as described above. Each tube for surface panels (Panel 1 - 12) was added with the 40 μL of reconstituted antibody mastermix and stained for 30 min. At the end of staining, each tube was washed with PBS and centrifuged for 5 min at 250 g, 4 °C. After removing supernatant by aspiration, cell pellets were resuspended with residual volume.

For fixation, permeabilization, and panel barcoding, Foxp3 Fixation/Permeabilization working solution and Permeabilization Buffer were prepared using FoxP3 Transcription Factor Staining Buffer set (eBioscience #00-5523) as manufacturer's instructions. At the fixation step, 0.5 mL of Foxp3 Fixation/Permeabilization working solution was added to each panel tube and vortexed briefly before incubation at room temperature for 1 h. Cells were washed with 0.5 ML of CSM and pelleted by centrifugation for 5 min at 600 g, 4 °C. After removing the supernatant by aspiration, each tube was barcoded with a unique combination of palladium isotopes[62].

For intracellular panel staining, intracellular panel was prepared as a single test mastermix in total 40 μl with Permeabilization Buffer. Cell pellets in each intracellular panel (Panel 13 - 15) were normalized to 60 μl with Permeabilization Buffer. Each panel mastermix was added to each tube and incubated for 45 min at room temperature. After incubation, cells were washed with CSM and centrifuged for 5 min at 600 g, 4 °C. Supernatant was removed by aspiration.

For core panel intracellular staining, intracellular staining portion of the core panel was prepared as a 15-tests mastermix in total 200 μL with Permeabilization Buffer. All samples were pooled into a single tube, washed with CSM, and centrifuged for 5 min at 600 g, 4 °C. Supernatant was removed by aspiration and resuspended with Permeabilization Buffer to a total volume of 300 μL. Intracellular core panel was added to the pooled sample, briefly vortexed, and incubated for 45 min at room temperature. After incubation, the sample was washed with CSM and centrifuged for 5 min at 600 g, 4 °C. After the supernatant was removed by aspiration, sample was briefly vortexed and mixed with 1 mL of DNA intercalator solution (1 mL of PBS supplemented with 100 μL of 16% PFA (Fisher 50980487), 0.5 μM Intercalator-Ir (Fluidigm 201192B) and 0.25 μM Intercalator-Rh (Fluidigm 201103 A)). Sample was incubated for 20 min at room temperature and then transferred to 4 °C for overnight storage before data acquisition.

## CyTOF staining for single panels
For CyTOF experiment for HSPC panel validation (Fig. 2D-Fig. 3) or surface marker identification (Fig. 5), 10x10e6 cells were thawed from liquid nitrogen, using the thawing media described above.

Each panel was prepared as two mastermixes, one for surface antibodies and one for intracellular antibodies. Mastermixes were filtered with Durapore 0.1 μm PVDF membrane filter for 2 min at 100 g to remove any possible precipitates. Before staining, BMMCs were suspended in CSM, added 1uL of TruStain FC Blocker (Biolegend 422302) per $10^6$ cells, and incubated for 10 min at room temperature. Subsequently, cells were stained with surface antibodies in CSM for 30 min at RT. All staining volumes were kept to 100 μL per 1 - $3 \times 10^6$ cells. After incubation, samples were normalized to a total of 1 mL with Low-Barium PBS and added 1 μL of 200 μM cisplatin (Sigma 232120) for labeling of non-viable cells for 5 min. Samples were washed in CSM and fixed using the Foxp3 Fixation/Permeabilization working solution for

                                                                                    

30 min. Intracellular staining of cells was done by adding antibodies using Permeabilization Buffer for 1 h. Prior to data acquisition, cells were stained with iridium DNA intercalator solution (1.6% PFA in low-barium PBS with 0.5 μM Ir191 intercalator for 20 min at RT or overnight at 4 °C.

## CyTOF sample acquisition

Before data acquisition, each sample was washed once with CSM and twice with ddH2O. Each wash step was followed by centrifugation for 5 min at 600 g, 4 °C. After the third wash, cells were resuspended with a 1:10 solution of EQ 4 element beads (Fluidigm 201078) in ddH2O to the concentration of 1e6 cells/mL and strained through a 35 μM FACS tube filter. Data was acquired on a CyTOF2 instrument (Fluidigm). Single cell events were recorded at a rate of ~500 cells/second.

## FACS sorting for inTAC-seq

For inTAC-seq, all mononuclear cells collected from two donors, each 50 ml of BM, were used for experiments. CD34[+] enriched samples were washed with PBS and stained with Live/Dead Aqua (Thermo Fisher L34966) as instructed by the manufacturer in dark for 20 min. Cells were washed with FBB buffer and spun down at 300 g, 5 min, 4 °C. Extracellular (E/C) panel (Supplementary Data 3) was added to samples in FACS buffer in the dark on ice for 30 min, followed by a wash with FACS buffer and a spin down at 300 g, 5 min, 4 °C. For fixation, each sample was fixed with 1 ml of 16% PFA for 1 min before washing with Permeabilization Buffer and centrifugation at 600 g, 5 min, 4 °C. Intracellular (I/C) panel (Supplementary Data 3) was prepared in Permeabilization Buffer and added to samples with a brief vortex. Intracellular staining was in the dark on ice for 30 min.

## FACS sorting for live cells

For FACS sorting for functional differentiation experiments (Fig. 6), 10x10e6 cells per donor were thawed from liquid nitrogen, using the thawing media described above. Thawed BMMCs were washed with FBB buffer and spun down at 250 g, 5 min, 4 °C. Primary panel (Supplementary Data 3) was added to samples in FACS buffer in the dark on ice for 30 min, followed by a wash with FACS buffer and a spin down at 300 g, 5 min, 4 °C. Secondary panel (Supplementary Data 3) was added to samples with a brief vortex, and incubated in the dark on ice for 30 min. At the end of incubation, PBS and Live/Dead Aqua were added to the sample and incubated for 20 more minutes in the dark at room temperature. Cells were sorted on a BD FACSAria Fusion (BD Biosciences) at the Stanford Shared FACS Facility.

## inTAC-seq sample processing and library preparation

ATAC-seq samples were prepared following Fast-ATAC protocol[44] with modification. Fixed, permed, sorted samples were spun down at 600 g for 5 mins and resuspended in resuspend in 15 μl of 1X TD Buffer supplemented with 0.1% NP40. And then added Tn5 in 1X TD Buffer. The amount of Tn5 was normalized to the cell number from the sorting (Donor 5: TdT[+] GMP 1,500 cells, TdT[-] GMP 14,000 cells, Donor 6: TdT[+] GMP 2,300 cells, TdT[-] GMPS, 15,000 cells). Cells were incubated at 37 °C with 1200 rpm shaking for 30 min. 2× reverse crosslinking buffer (2% SDS, 0.2 mg/mL proteinase K, and 100 mM N,N-Dimethylethylenediamine, pH 6.5 [Sigma Aldrich D158003]) was added at equal volume to transposed cells and reversal of crosslinks was performed at 37 °C overnight with 600 rpm shaking. DNA was purified using Qiagen minelute PCR purification columns (Qiagen 28006). Following purification, library fragments were amplified with NEBnext PCR master mix (NEB M0541S) and 1.25 μM of Nextera PCR primers, using the following PCR conditions: 72 °C for 5 min; 98 °C for 30 s; and thermocycling at 98 °C for 10 s, 63 °C for 30 s and 72 °C for 1 min. After first five cycles, a 5 μl aliquot of the PCR reaction was added 10 μl of the PCR cocktail with Sybr Green (Invitrogen S7563) at a final concentration of

0.6×. Side qPCR reaction was carried out for 20 cycles to determine the additional number of cycles needed for the remaining 45-μL reaction. The libraries were purified using a Qiagen PCR cleanup kit (Qiagen 28104)[63].

## OP9 and OP9-DL4 maintenance

OP9 cells and human DL4 expressing OP9-DL4 cells[50] were gifted from Zúñiga-Pflücker lab. Cells were cultured in a 100 mm-dish using freshly prepared OP9 media (MEM α, no nucleosides (Gibco12561056) supplemented with 15% FBS and 100 units/mL of Penicillin and 100 μg/mL). Cells were split 1:4 or 1:5 when reached 90% confluency[49].

## OP9-DL4 bulk co-culture differentiation assay for T, NK, Myeloid cell differentiation

1 confluent 100 mm OP9-DL4 plate was divided into 2 6-well plates and incubated overnight in a 37 °C, 5% CO$_2$ incubator. 24 hr after plating OP9-DL4 cells, 600 cells from each sorted HSPC population were deposited into each well and were cultured in the presence of 10 ng/ml SCF (PeproTech 10780-454), 5 ng/ml FLT3L (PeproTech 10773-618) and 5 ng/ml IL-7 (PeproTech 200-07) in OP9 media (SF7 media). Cells were dissociated from wells by pipetting, filtered with 70 μm cell strainer, centrifuged at 300 g for 5 min and transferred to new plates with fresh OP9-DL4 weekly. Harvested cells were analyzed by flow cytometry at weeks 3 and 5 (Flow panel in Supplementary Data 3). Entire volume in each well was analyzed and wells with fewer than 5 human cells (live hCD45[+]) were excluded from analysis.

## OP9 bulk co-culture differentiation assay for B cell differentiation

1 confluent 100 mm OP9 plate was divided into 2 6-well plates and incubated overnight in a 37 °C, 5% CO$_2$ incubator. 24 hr after plating OP9 cells, 500 cells from each sorted HSPC population were deposited into each well and were cultured in the presence of 10 ng/ml IL-7 in OP9 media (SF7 media). Cells were dissociated from wells by pipetting, filtered with 70 μm cell strainer, centrifuged at 300 g for 5 min and transferred to new plates with fresh OP9 after a week. After 2 weeks, cells were harvested and analyzed by flow cytometry (Flow panel in Supplementary Data 3). Entire volume in each well was analyzed and wells with fewer than 5 human cells (live hCD45[+]) were excluded from analysis.

## OP9-DL4 limiting dilution assay

OP9-DL4 cells were plated at a concentration of 2,500 cells in 100 μl per well in a flat-bottom tissue culture treated 96-well. 24 hr after plating OP9-DL4 cells, different HSPC populations were sorted directly onto 96-well plates. After sorting, 100 μl of 2× SF7 media was added in each well. Half of the media was replaced every week. After 2.5 weeks of culture, cells were dissociated by pipetting and transferred to v-bottom 96-well for staining and flow cytometry analysis (Flow panel in Supplementary Data 3). Entire volume in each well was analyzed and wells.

## OP9-DL4 single-cell cloning assay

OP9-DL4 cells were plated at a concentration of 2,500 cells in 100 μl per well in a flat-bottom tissue culture treated 96-well. 24 hr after plating OP9-DL4 cells, a single CD84[lo] GMP was sorted onto 96-well plates. After sorting, 100 μl of 2× SF7 media was added in each well. Half of the media was replaced every week. After 2 weeks of culture, cells were dissociated by pipetting and transferred to v-bottom 96-well for staining and flow cytometry analysis (Flow panel in Supplementary Data 3). Entire volume in each well was analyzed and wells.

## Methylcellulose colony formation assay

MethoCultTM (StemCell Technologies, H4435) was used as the manufacturer's instruction. Briefly, frozen aliquots of MethoCultTM

were thawed overnight. Sorted HSPC populations were diluted with IMDM with 2% FBS to 10X of desired final concentration for seeding. 300 μl of diluted cells were mixed to 3 ml of MethoCultTM and vortexed thoroughly and then incubated for 5 min to reduce bubbles. MethoCultTM mixture containing cells were drawn with a sterile 16-gauge Blunt-End Needle to a sterile 3 ml syringe. 1.1 ml of the mixture was slowly distributed to a well in 6-well SmartDishTM plates (StemCell Technologies, 27370). 6-well plates were incubated in a 37 °C, 5% $CO_2$ incubator for 2 weeks. Differentiation results were counted via STEMvisionTM (StemCell Technologies, 22006) with Color Human BM 14-Day software.

### Quantification and statistical analysis

**CyTOF data preprocessing.** Acquired samples were bead normalized using MATLAB based normalization software[64]. Sample debarcoding was performed using the premessa R package. Normalized and debarcoded data was then uploaded to either the Cytobank analysis platform (https://www.cytobank.org) or the Cell Engine analysis platform (https://www.cellengine.com). Gated data was downloaded and further analyzed using the R programming language (https://www.r-project.org) and where applicable, Bioconductor (https://www.bioconductor.org) software. Data was transformed using standard inverse hyperbolic sine (asinh) transformation with a co-factor of 5 and column normalized for each individual marker.

**Proteomic screen data integration and clustering.** To integrate the data collected on 15 different panels, we first clustered cells into 400 FlowSOM clusters[65] (median cell number per cluster: 1288) using the conserved panel. Then each FlowSOM cluster was given the median value for each target in the screen. For meta-clustering, we generated a nearest-neighbor graph of FlowSOM clusters using all median values of 81 targets and then performed Leiden-clustering using FindNeighbors and FindClusters functions, respectively, from Seurat package (https://satijalab.org/seurat).

**Differential protein expression analysis.** Differential analysis of protein markers between meta-clusters were performed[21]. Differences in the distribution of molecules were calculated in equally subsampled populations using the KS test. P values were considered significant if the Bonferroni corrected p value < 0.05 to prevent inclusion of false positives in the comparisons.

**ATAC-seq data processing.** Adapter sequence trimming, mapping to the human (hg38 or hg19) reference genome using Bowtie2 and PCR duplicate removal using Picard Tools were performed. hg38 was primarily used for analysis, but the raw data was re-mapped to hg19 genome when comparing to the public dataset mapped to hg19 genome. Mitochondrial reads mapping to chrM were removed from downstream analysis. Preprocessed bam files were loaded into R using DsATAC.bam function in the ChrAccR R package (https://greenleaflab.github.io/ChrAccR/index.html). To create a consensus peakset across technical and biological replicates, getPeakSet.snakeATAC function in the ChrAccR package was used.

**Differential accessibility analysis.** Differential peaks in the consensus peakset were called by DESeq2 via createReport_differential function in ChrAccR package. Transcription factor motifs enrichment scores in the differential peaks were calculated by ChromVAR package[45] via createReport_explanatory function in ChrAccR package.

**inTAC-seq Projection onto scATAC UMAP space.** inTAC-seq sample prepared in bulk was projected onto scATAC UMAP space[44]. Processed bam files from inTAC-seq samples were downsampled to approximately 150 cells per sample, and loaded into R using DsATAC.bam function in the ChrAccR R package. Count matrix for 500 bp tiling regions was converted into a summarizedExperiment data class. Pseudo-single-cells were simulated by subsampling from the summarizedExperiment data class and each pseudo-single-cell was calculated for its UMAP coordinates based on calculating iterativeLSI by using the projectBulkATAC function from ArchR package[66].

**Active Progenitor Frequency Estimation from Limiting Dilution Assay.** ELDA software (https://bioinf.wehi.edu.au/software/elda/)[53] was used for statistical analysis of the limiting dilution assay results.

### Reporting summary

Further information on research design is available in the Nature Portfolio Reporting Summary linked to this article.

## Data availability

The mass cytometry fcs files and the normalized count table for the screen generated in this study have been deposited in Dryad at https://doi.org/10.5061/dryad.xgxd254jp. The finalized mass cytometry panel fcs files, inTAC-seq bam files, and co-culture fcs files in the manuscript are available at https://doi.org/10.5061/dryad.1c59zw3zt. The processed data for generating figures are provided in the Source Data file. All other data are available in the article and its Supplementary files or from the corresponding author upon request. Source data are provided with this paper.

## Code availability

The code for processing and analyzing the mass cytometry the screen is available at https://doi.org/10.5061/dryad.xgxd254jp.

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

## Acknowledgements

We thank members of the Bendall and Greenleaf labs for their support and advice. This study was supported by grants from National Institutes of Health 1DP2OD022550-01, 1R01AG056287-01, 1R01AG057915-01, R01AG068279, UH3CA246633 and 1U24CA224309-01 (to S.C.B), RM1-HG007735, UM1-HG009442, UM1-HG009436, 1UM1-HG009442, U2CCA233311, U54-GH010426, and U19-AI057266 (to W.J.G.). This work is also supported by the Defense Advanced Research Project Agency (W911NF1920185 to W.J.G.) and a Stanford Cancer Institute-Goldman Sachs Foundation Cancer Research Award (to W.J.G). W.J.G. is a Chan Zuckerberg investigator. Y.K. is supported by Stanford Immunology Baker Fellowship and by KFAS Overseas PhD Scholarship from Korea Foundation for Advanced Studies. A.A.C. was supported by the NIAID of the National Institutes of Health under award number 5T32AI007290-32 and the National Science Foundation Graduate Research Fellowship Program under grant number DGE-1656518. A.T. was supported by the Damon Runyon Cancer Research Foundation (DRG-118-16).

## Author contributions

Y.K., A.A.C., W.J.G., and S.C.B. conceived and designed the study. Y.K., A.A.C., P.F., D.H., and L.B. performed experiments. D.R.G. and A.G.T. generated critical reagents. D.R.G. provided scripts for data processing. Y.K. performed data analysis and wrote the manuscript. W.J.G. and S.C.B supervised and funded the study.

## Competing interests

W.J.G. is a consultant for 10XGenomics, Guardant Health, Quantapore, Erudio Bio. and Lamar Health and cofounder of Protillion Biosciences and is named on patents describing ATAC-seq. The other authors declare no competing interests.
