## [Peer Review File · Nature Communications]

Terminal deoxynucleotidyl transferase and CD84 identify human multi-potent lymphoid progenitorsREVIEWER COMMENTS

Reviewer #1 (Remarks to the Author):

The paper by Kim et al take advantage of a CyTOF based dataset for the identification of cellular subsets in human blood cell development. Their analysis identifies a TdT+ population within the classical GMP compartment. Using inTAC-seq the authors identify differentially accessible regions enriched for binding sites for E- and IRF proteins. The authors find that CD84 low cells are enriched for TdT expressing cells and explore this finding to determine that the T-lineage potential within the GMP compartment is detected within the CD38 low population.

The paper is well written and based on a logical set of experiments. My major concern is well formulated by the authors on row 397 in the discussion, the data presented does not resolve if the apparent bipotentiality is due to a shared multipotent progenitor or a consequence of a mixed population of unipotent cells. Without such information the paper leaves us with the conclusion that cells with T-lineage potential can be found in the CD84 low GMP population. This is interesting but does not dramatically develop our understanding of developmental trajectories in human hematopoiesis.

Specific comments:

1: I do not believe that the data support the title as no functional link between TdT expression and lymphoid lineage potential is established. The authors only examine the lineage potential of CD84^{low} cells among which only about one third are TdT positive and the molecular analysis does not make a strong case for lymphoid priming.

2: The authors use the term lymphoid potential in the title but they only examine T/NK-lineage potential. Determination of the B-lineage potential of both CD84^{low} and CD84^{high} cells would provide more conclusive evidence to this end.

3: The authors could consider to use SC-RNA seq data to identify the TdT positive cells within the GMP compartment. This should allow the authors to better understand the nature of the priming in these cells and possibly identify a more useful surface marker to

enrich for TdT positive cells. The absolute majority of the CD84^{low} cells are clearly TdT negative.

4: Even though the inTAC-seq provides interesting data, it should be possible to identify these cells analyzing scATAC data to obtain a better picture of the epigenetic landscape in the lymphoid primed cells.

5: The authors use limiting dilution to study the potential of the cells. However, as the cells may have differential proliferative capacity, single cell cloning assay might provide more conclusive evidence. These assays could also be modified to resolve the issue of bipotentiality in the CD84^{low} population.

Minor comments:

1: In mouse B-cell development, the expression of Cd179b has been reported as is restricted to B-lineage committed cells (Jensen et al JEM 2018). The data present in this report would suggest that this may not be true for human lymphoid development. This could be worth mentioning in the discussion.

2: I have a hard time to identify support for that ID4 binds DNA. The authors may wish to explain how such a binding motif is identified.

Reviewer #2 (Remarks to the Author):

The manuscript « Expression of terminal deoxynucleotidyl transferase (TdT) identifies lymphoid-primed progenitors in human bone marrow » describes a multi-omics analysis of human bone marrow from healthy adult donors. They identify a population that corresponds in phenotype to GMP but expresses TdT, an enzyme used in VDJ recombination of B and T cell progenitors. The authors show that this subset of GMP clusters with LMPP, show epigenetic signature of a lymphoid progenitor and can differentiate into T lineage cells in vitro.

Although the manuscript has a strong potential a better characterization is required.

Major concerns.

1. CD84 is used as a surrogate to TdT expression. However, from the plots shown, although most TdT expressing cells fall in the CD84 low gate there are many cells that fall in the same gate and are TdT negative (Figure 5b), so it is a bad surrogate.
2. The TdT + GMP subset is described as lymphoid primed by epigenetic analysis. Although it is clearly different from TdT- GMP the reader would like to know does it compare with LMPP in similar assay? The assays done here do not allow distinction between primed, committed or simply multipotent.
3. That these cells differentiate into T lineage cells in vitro with similar frequency to LMPP is only telling that they have T cell potential and not that they are biased.
4. It is unclear if the authors think these cells retain B and NK cell potential. It is also unclear why these potentials were not tested.
5. Are the cells that generate myeloid cells also capable to differentiate into lymphocytes?

At the end it is unclear whether the properties described here are seen in all CD84- cells or only the TdT expressing subset.

Reviewer #3 (Remarks to the Author):

The authors assess heterogeneity within human CD34+ hematopoietic stem and progenitor cells (HSPCs) using a highly multiplexed single cell proteomic approach. Given the unclarity about early cell fate decisions in human HSPCs, this is a study of outstanding interest in the field of hematopoiesis/HSPC biology/immunology.

The data acquisition is very robust and well controlled. While very decent numbers of biological replicates are used, the assessment of controls for the mass cytometry approach as well its bioinformatic analysis is outside the area of expertise of this referee.

To zoom into HSCPCs, the authors identify the proteomic signature of CD34- cells and report a bifurcation between cells with lymphoid and myeloid potential compared to erythroid and megakaryocytic potential and find the bifurcation confirmed in transcriptomic and epigenetic landscapes. This is of outstanding interest. This part of the study would very

much benefit from validation of this claim through the establishment of a cell surface-based prospective isolation protocol of key populations complemented by functional validation of this claim. Such validation could be run through comparable assays as the authors have used for TdT+ GMPs in the manuscript.

Based on protein expression patterns, the authors identify a TdT+ cell cluster that they allocate between CD38^{lo} LMPPs and CD10+ progenitors. Mapping of sc-proteome data to canonical HSPC identification revealed unanticipated heterogeneity within the lympho-myeloid differentiation axis with clear bimodal division of the canonical GMP population into TdT+ vs TdT- cells. Immunophenotypical TdT+ GMPs are termed 'previously unappreciated lymphoid progenitors', based on a lymphoid-primed chromatin accessibility landscape that was determined using self-generated and available data sets. Next, the authors show that the expression of CD84 inversely correlates with TdT positive GMPs and enriches for putative lymphoid progenitors within GMPs. The authors confirm convincingly the distinct progenitor potential using suitable stromal layer-based and colony assays.

minors

This referee is surprised about the sharpness of the statement of the authors in the opening paragraph of the introduction on the concept of stepwise differentiation stages of hematopoietic progenitors. This is perceived as unnecessary by this referee because this was a very suitable model based on the methodologies available for a long time and it was interpreted very widely, thus, plasticity during differentiation stages was acknowledged and assumed, and therefore not treated as an 'ideal concept of progenitor cell types'. This referee recommends rephrasing, which has no effect on the importance of the current study.

The authors are asked to include information on the specificity of the antibodies used in each dot plot of Fig S7a.

Reviewer #4 (Remarks to the Author):

The study submitted by Kim et al. is an exciting tour-de-force, using a high throughput

proteomic screen of human bone marrow progenitors to identify a subset of TdT+ GMPs with mixed lymphoid and myeloid potential, with these cells enriched in the CD84-low subset of GMPs. These data are therefore important for the field, elucidating new complexities in lympho-myeloid differentiation. Furthermore, these studies are performed with human progenitors and identify cell surface molecules which can be used to isolate live cells.

Specific comments:

I may have missed this information but it wasn't clear to this reviewer as to the percentages of cells that encompass each cluster from A1 to A10. It might also be interesting to present the cluster map as a function of the cell cycle phase. From the data in Fig 5C/D, it appears that approximately 1% of all CD34+ cells are TdT-high GMPs or 8% of total GMPs. Is this correct? This information could be presented in the manuscript (and compared with information from previous publications in lines 403-404).

From a technical perspective, as relates to diversity between samples, the authors indicate that they used BM from 8 healthy adult donors. It would be helpful to indicate the age range of these donors. Additionally, in some of the figures (i.e. fig 2, 3, 5), it wasn't clear as to how many BM donors were used for the analyses. While outside the scope of this study, it would be of interest to assess whether these clusters are present at different levels in young vs old donors (as well as CD84-Hi vs CD84-Lo GMP). The authors may want to comment on this in the discussion.

The identification of a TdT+ GMP is a very interesting and novel aspect of this manuscript. In this regard, what is the half-life of TdT? If the half-life is very long, this might result in maintained expression. That being said, the data in Fig. 3B show levels of TdT in GMP that are equivalent to that in CLP. Are mRNA levels similar in these two subsets ((ie. CD84-low GMP vs CLP)?

The authors state that they focused on T lymphoid potentials because "B lymphoid cell origins in humans have been well documented (line 374)." Nonetheless, it is interesting to note that a murine progenitor population in the thymus with T and myeloid potential, but

not B lymphoid potential, has been described by 2 groups (10.1038/nature06840, 10.1038/nature06839). As such, it would be interesting to determine whether CD84-Low GMP possess lymphoid activity and at the least, to discuss their data in comparison with this murine population.

Minor

The authors state that they “identified CD84^{lo} as a surrogate for TdT⁺ GMPs for live-cell sorting (line 341).” While their sorting resulted in an impressive 40% of TdT⁺ cells, this reflects an important enrichment but not an identification. Conversely though, the authors clearly show that TdT⁺ GMP are not present in the CD84-high subset. The wording of this phrase can be modified.

Reviewer #5 (Remarks to the Author):

Kim et al use high dimensional immune profiling to identify TdT⁻ CD84^{lo} granulocyte-monocyte progenitors (GMP) as upstream of lympho-myeloid progenitors. Altogether this is a very rigorous piece of work with sound conclusions.

Minor:

1. Please provide information on the number events and, or range acquired and comment regarding confidence in identifying specific subsets.
2. I found the figures legends too brief. I was unable to interpret the figures independent of the main text.
3. The main text is very heavy reading. The work would benefit from a few explanatory sentences to help readers who are less expert in high dimensional analysis. Similarly, there are many long complex sentences that could be split to make the text more intelligible

Reviewer #1 (Remarks to the Author):

The paper by Kim et al take advantage of a CyTOF based dataset for the identification of cellular subsets in human blood cell development. Their analysis identifies a TdT+ population within the classical GMP compartment. Using inTAC-seq the authors identify differentially accessible regions enriched for binding sites for E- and IRF proteins. The authors find that CD84 low cells are enriched for TdT expressing cells and explore this finding to determine that the T-lineage potential within the GMP compartment is detected within the CD38 low population.

The paper is well written and based on a logical set of experiments. My major concern is well formulated by the authors on row 397 in the discussion, the data presented does not resolve if the apparent bipotentiality is due to a shared multipotent progenitor or a consequence of a mixed population of unipotent cells. Without such information the paper leaves us with the conclusion that cells with T-lineage potential can be found in the CD84 low GMP population. This is interesting but does not dramatically develop our understanding of developmental trajectories in human hematopoiesis.

We appreciate the reviewer's feedback on the clarity and messaging in the manuscript. As part of the revision, we have performed significant new experiments including clonal assessment of multi-lineage output and deeper phenotyping that we believe address the reviewer's suggestions (*below*).

Specific comments:

1: I do not believe that the data support the title as no functional link between TdT expression and lymphoid lineage potential is established. The authors only examine the lineage potential of CD84low cells among which only about one third are TdT positive and the molecular analysis does not make a strong case for lymphoid priming.

With respect to epigenetic priming, it is defined as the 'pre-patterning' of chromatin landscape that occurs toward a specific lineage prior to locking in the cell fate (Chen & Dent, Nature Reviews Genetics, 2014). In this manuscript, we identified TdT+ progenitors that already express lymphoid-specific proteins and whose open chromatin regions enrich for lymphoid-specific transcription factors' motifs and chromatin accessibility landscape distinctly closer to other lymphoid progenitors. Thus, we conclude that these progenitors are 'primed' for lymphoid lineage.

Furthermore, we agree with the reviewer that the limitation of this paper is the lack of prospective isolation between TdT+ cells and CD84lo TdT- cells. However, TdT expression (which we require ~100+ copies per cell to see) may be sufficient, but not necessary, as even the TdT- CD84lo cells fall in the 'Cluster1 lymphomyeloid progenitor' in high dimensional space (Right) [Figure 5E, revised manuscript]. This is still in line with our goal of identifying the missing lymphoid potential.

In other words, we have utilized TdT as the concrete target to identify lymphoid progenitors, and our follow up experiments successfully found the link between TdT expression and CD84lo phenotype, discovering even more previously misidentified lymphoid progenitors. Hence, our title highlights the importance of TdT as the molecular guide to identify lymphoid progenitors, because if not for TdT, we would not have been

able to define the lymphoid progenitors in first place. At the same time, in the abstract, we articulate that we measured the lymphoid potentials with CD84^{lo} cells, to clearly deliver our findings.

2: The authors use the term lymphoid potential in the title but they only examine T/NK-lineage potential. Determination of the B-lineage potential of both CD84^{low} and CD84^{high} cells would provide more conclusive evidence to this end.

We thank the reviewer for the suggestion. We performed additional experiments and have added B cell differentiation assay data in revised Figure 6D and updated the manuscript and attached the edited part below. These cells also possess this lineage potential.

[Revised manuscript lines 315-321]

To confirm the all-lymphoid potentials, we used OP9 co-culture system to measure B cell differentiation potentials. CD84^{lo} cells robustly proliferated in OP9 co-culture media in the presence of IL-7 and developed into CD19⁺ B cells. On the other hand, CD84^{hi} cells could not yield any B cells. At the same time, CD10⁺ CLPs were less proliferative but more effectively differentiated into CD19⁺ B cells. This is consistent with our cluster identification of CD84^{lo} cells as earlier lymphoid progenitors, CD10⁺ CLPs as B cell progenitors, and CD84^{hi} cells as myeloid progenitors devoid of lymphoid potentials.

[Revised Figure 6D]

(D) Results of B cell differentiation from OP9 bulk coculture in week 2. Overall proliferation (left) and B lineage output (right).

3: The authors could consider using SC-RNA seq data to identify the TdT positive cells within the GMP compartment. This should allow the authors to better understand the nature of the priming in these cells and possibly identify a more useful surface marker to enrich for TdT positive cells. The absolute majority of the CD84^{low} cells are clearly TdT negative.

While we agree with the reviewer that scRNA-seq analysis with an identification of TdT⁺ GMP cells would be interesting, there is no robust, available method to stain for intracellular proteins with single-cell RNA-seq in hematopoietic cells up to date. Recent preprint by Blair et al. (biorxiv, 2023) states that they could not achieve decent RNA quality with intracellular staining and decided to couple ATAC-seq with intracellular staining, which is what our iTAC-seq data here already achieves.

Even so, to maximize the datasets we already have, we have performed additional analysis of healthy donors' scRNA-seq data from our previous publication (Granja, Klemm, McGinnis et al., Nat

Biotech. 2019). In the violin plots below, we have analyzed the RNA level of TdT (gene name DNNT) and CD10 (gene name MME) by the clusters annotated by the authors of the paper.

Most of TdT (gene name: DNNT) expressing cells (Cluster 6 CLP.1, Cluster 15, CLP.2) among CD34+ progenitors are CD10 (gene name: MME) positive as well. While we observe some cells in Cluster 5 CMP_LMPP and Cluster 7 GMP with slight upregulation of TdT, their RNA transcriptome is not distinct enough from the rest of Cluster 5 CMP_LMPP cells to give specific cell surface marker candidates. We suspect the protein level upregulation of TdT among CD10- cells is a very rapid process that cannot be captured with the sensitivity of scRNA-seq – highlighted here by the Poisson distribution of gene counts here.

Furthermore, we have revisited the scATAC-seq dataset that was used to compare our inTAC-seq data and looked for marker genes for TdT+ cells, With the threshold of FDR ≤ 0.05 and log2 fold change ≥ 1 , we identified 9 additional marker genes (MEIS1, CALN1, FAM30A, SCN8A, COL5A1, GPR12, C20orf203, SHANK3, NPTX2) - but none were annotated to be cell surface proteins. These analyses have been updated in the revised manuscript.

[Revised manuscript lines 282-284]

Of note, we have analyzed inTAC-seq data with reference scATAC-seq data to identify TdT+ GMP marker genes (with threshold of FDR ≤ 0.05 and log2 fold change ≥ 1), but none of the identified marker genes was cell surface protein.

Thus, while we agree with the reviewer that CD84lo not the ideal surface marker surrogate, it is the best we can find with current state-of-the-art technologies. We want to emphasize our proteomic screen spanning 351 surface proteins likely exhausted readily available flow cytometry antibodies that are practical candidates. Furthermore, our proteomic analysis of TdT- CD84lo cells showed that these cells overlap with Cluster 1 Early lympho-myeloid progenitors, and CD84hi cells are clearly all TdT- and overlap with Cluster 4 Myeloid progenitors. Hence, our CD84lo surface phenotype definition, both by proteomics analyses and functional differentiation assays, represents lymphoid progenitors that have been misidentified as GMPs.

4: Even though the inTAC-seq provides interesting data, it should be possible to identify these cells analyzing scATAC data to obtain a better picture of the epigenetic landscape in the lymphoid primed cells.

We have compared our TdT+ cells to the scATAC-seq data from our previous publication (Granja, Klemm, McGinnis et al., Nat Biotech. 2019) in Figure 4E. This analysis revealed that the TdT+ cells

have the chromatin accessibility of 'LMPP's, which in the original publication have been annotated by their overall epigenetic landscape, not the surface phenotypes.

[Figure 4E, revised manuscript]

E Projection onto Reference Single Cell UMAP

5: The authors use limiting dilution to study the potential of the cells. However, as the cells may have differential proliferative capacity, single cell cloning assay might provide more conclusive evidence. These assays could also be modified to resolve the issue of bipotentiality in the CD84^{low} population.

We thank the reviewer for their suggestion. While particularly challenging due to the technical constraints of growing out single progenitor cells for lineage phenotyping, we were able to get a clonal assay running after much optimization. In the revised manuscript we show single-cell cloning assays with the OP9-DL4 coculture system and analyzed the bipotentiality of CD84^{lo} cells. We observed substantial bipotentiality in these cells (both uni- and bi-lineage potential). We have revised our manuscript and Figure 6 and attached the updated sections below.

[Revised Figure 6G]

G Single-cell differentiation

(G) Lineage potentials from single-cell cloning assay of CD84^{lo} cells. Less than three CD7⁺ cells (lymphoid) or CD14⁺ or CD15⁺ cells (myeloid) were classified as N/A.

[Revised manuscript lines 342-354]

Lastly, to assess whether the CD84^{lo} cells are lympho-myeloid bipotential progenitors or mixture of unilineage lymphoid progenitors and myeloid progenitors, we conducted a single-cell cloning assay of CD84^{lo} cells with an additional donor bone marrow (Donor 9). We note that SGF15/2 culture⁶ for multi-lineage readout was not successful with adult bone marrow HSPCs in our hands. Instead, we optimized OP9-DL4 coculture system to measure CD7⁺ lymphoid cells and CD14⁺ or CD15⁺

myeloid cells. Out of 132 wells cloned, 28.0% (37 wells /132 wells) were successfully cloned. Of the wells positively cloned, 21.6% had both lymphoid and myeloid progeny, and the other 21.6% and 48.6% had lymphoid only and myeloid only progeny, respectively (Figure 6G). Thus, we report a substantial lympho-myeloid bipotentiality at clonal level among CD84^{lo} cells. Compared to a previous study using human cord blood samples⁶, this level of bipotentiality is only seen among LMPPs, corroborating our annotation of these cells as early lympho-myeloid progenitors in proteomic space.

Minor comments:

1: In mouse B-cell development, the expression of Cd179b has been reported as is restricted to B-lineage committed cells (Jensen et al JEM 2018). The data present in this report would suggest that this may not be true for human lymphoid development. This could be worth mentioning in the discussion.

We thank the reviewer for the feedback. We have updated the manuscript about CD179b expression and included the associated references.

[Revised manuscript lines 200-204]

Interestingly, while we detect intracellular CD179b (Gene name: IGLL1) protein expression in Cluster 1 Ly-My progenitors and Cluster 6 Lymphoid progenitors, before Cluster 8 B committed progenitors, the mouse homolog *Igll1* has been shown to be restricted in B committed progenitors³⁸⁻⁴⁰. This observation suggests possible differences in gene regulation between humans and mice in lymphoid development.

2: I have a hard time to identify support for that ID4 binds DNA. The authors may wish to explain how such a binding motif is identified.

For computational analysis, we have used the JASPAR motifs in the database (<https://jaspar.genereg.net/>), and the data suggests that the ID4 motif is enriched in the DNA. According to the JASPAR database, the motifs in JASPAR can be collected either internally generated motifs, by analyzing ChIP-seq/-exo sequences using a custom motif discovery pipeline or externally taken directly from other publications and/or resources (<https://jaspar.genereg.net/>).

Reviewer #2 (Remarks to the Author):

The manuscript « Expression of terminal deoxynucleotidyl transferase (TdT) identifies lymphoid-primed progenitors in human bone marrow » describes a multi-omics analysis of human bone marrow from healthy adult donors. They identify a population that corresponds in phenotype to GMP but expresses TdT, an enzyme used in VDJ recombination of B and T cell progenitors. The authors show that this subset of GMP clusters with LMPP, show epigenetic signature of a lymphoid progenitor and can differentiate into T lineage cells in vitro.

Although the manuscript has a strong potential a better characterization is required.

Major concerns.

1. CD84 is used as a surrogate to TdT expression. However, from the plots shown, although most TdT expressing cells fall in the CD84 low gate there are many cells that fall in the same gate and are TdT negative (Figure 5b), so it is a bad surrogate.

We agree with the reviewer that CD84^{lo} phenotype is not the ideal or identical substitute for TdT expression. And as the reviewer pointed out, there are substantial number of TdT- CD84^{lo} cells. However, as we responded to Reviewer 1 (Point 1), TdT expression (which we require ~100+ copies per cell to see) may be sufficient, but not necessary, as even the TdT- CD84^{lo} cells fall in the 'Cluster1 lympho-myeloid progenitor' in high dimensional space (revised Figure 5E, attached below). This is still in line with broadly identifying the missing lymphoid potential. This has been noted in the revised manuscript.

[Figure 5E, revised manuscript]

Furthermore, our screen of 351 surface molecules likely exhausted publicly available flow-ready antibodies. Thus, while not ideal, CD84 is currently best marker for the field as of now. (also see Reviewer 1, point 3)

2. The TdT + GMP subset is described as lymphoid primed by epigenetic analysis. Although it is clearly different from TdT- GMP the reader would like to know does it compare with LMPP in similar assay? The assays done here do not allow distinction between primed, committed or simply multipotent.

We agree with the reviewer that the comparison with LMPP is an important aspect. In the revised manuscript, we have compared TdT+ and TdT- GMPs to LMPP on their epigenetic landscapes with sorted bulk population in Figure 4D and reference single-cell ATAC-seq annotation in Figure 4E. In both analyses, TdT+ GMP are closer to LMPPs in bulk analysis (Revised Figure 4D) and at least partially overlapping with LMPPs in single-cell analysis (Revised Figure 4E). Therefore it seems they cells serve as the natural developmental bridge.

[Revised Figure 4D]

[Revised Figure 4E]

As we responded to Reviewer 1 (Point 1), epigenetic priming is defined as the 'pre-patterning' of chromatin landscape that occurs toward a specific lineage prior to locking in the cell fate (Chen & Dent, Nature Reviews Genetics, 2014). In this manuscript, we identified TdT+ progenitors that already express lymphoid-specific proteins and whose open chromatin regions enrich for lymphoid-specific transcription factors' motifs and chromatin accessibility landscape distinctly closer to other lymphoid progenitors. Thus, we conclude that these progenitors are 'primed' for lymphoid lineage.

3. That these cells differentiate into T lineage cells in vitro with similar frequency to LMPP is only telling that they have T cell potential and not that they are biased.

We agree that the *in vitro* differentiation result of CD84^{lo} GMPs itself do not prove the T cell bias in these cells. Nevertheless, LMPP is the first lymphoid-primed cell type in the bone marrow hematopoiesis hierarchy and has been reported to be the most efficient in T cell development (Ghaedi et al., Cell Reports, 2016). Therefore, the fact that CD84^{lo} cells were as efficient as LMPPs implies that we have discovered a different source of highly effective T cell precursor population in the bone marrow.

Furthermore, as we responded above, our definition of lymphoid priming is rooted in the definition of epigenetic priming. The lymphoid-specific chromatin accessibility and protein expression patterns suggest lymphoid priming of these cells, compared to their mistaken myeloid-committed identity as GMPs. Given this, we believe our interpretation of the results represents a more accurate model of human lymphoid development.

4. It is unclear if the authors think these cells retain B and NK cell potential. It is also unclear why these potentials were not tested.

In our revised manuscript, we present data from new experiments where NK cell potential and B cell potential were measured in revised Figure 6B and 6D, respectively. The related subpanels are marked in red rectangles below. As expected these cells possessed this potential similar to what would be expected based on the original T cell outputs.

[Revised Figure 6A~D]

5. Are the cells that generate myeloid cells also capable to differentiate into lymphocytes?

We thank the reviewer for their suggestion. As mentioned in our response to Reviewer 1 (Point 5), We have conducted single-cell cloning assays for the revised manuscript with the OP9-DL4 coculture system and analyzed the bipotentiality of CD84lo cells. We observed substantial lymphoid-myeloid bipotentiality in these cells. We have revised our manuscript and Figure 6 and attached the updated sections below.

[Revised Figure 6G]

G Single-cell differentiation

(G) Bar plot of the single-cell cloning assay results. A single CD84lo GMP on OP9-DL4 mono-layer was cultured for 2 weeks and then measured for the existence of lymphoid (CD7+) cells or myeloid (CD14+ or CD15+) cells. If neither lineage had three or more cells, the well was marked as not applicable (NA) for lineage potentials.

[Revised manuscript line 339-351]

Lastly, to assess whether the CD84lo cells are lympho-myeloid bipotential progenitors or mixture of unilineage lymphoid progenitors and myeloid progenitors, we conducted a single-cell cloning assay of CD84lo cells with an additional donor bone marrow (Donor 9). We note that SGF15/2 culture⁶ for multi-lineage readout was not successful with adult bone marrow HSPCs in our hands. Instead, we optimized OP9-DL4 coculture system to measure CD7+ lymphoid cells and CD14+ or CD15+ myeloid cells. Out of 132 wells cloned, 28.0% (37 wells /132 wells) were successfully cloned. Of the wells positively cloned, 21.6% had both lymphoid and myeloid progeny, and the other 21.6% and 48.6% had lymphoid only and myeloid only progeny, respectively (Figure 6G). Thus, we report a substantial lympho-myeloid bipotentiality at clonal level among CD84lo cells. Compared to a previous study using human cord blood samples⁶, this level of bipotentiality is only seen among LMPPs, corroborating our annotation of these cells as early lympho-myeloid progenitors in proteomic space.

At the end it is unclear whether the properties described here are seen in all CD84- cells or only the TdT expressing subset.

We agree with the reviewer that the limitation of this paper is the lack of prospective isolation between TdT+ cells and CD84lo TdT- cells. However, as mentioned in our response to Reviewer 2's point 1, CD84lo cells are the best currently available phenotype to enrich TdT+ cells. And as the high-dimensional proteomic data in Fig 5E (attached below) indicate TdT- CD84lo cells to be early lympho-myeloid progenitors, we suggest that overall CD84lo GMPs, unlike CD84hi GMPs, should be considered as a population with lymphoid potential.

[Figure 5E, revised manuscript]

E CD84lo and CD84hi GMPs UMAP projection

Thus, while the surface phenotype to distinguish CD84lo TdT+ cells and CD84lo TdT- cells remains unknown, we argue that it is minor to the importance of identifying novel lymphoid progenitors in this manuscript as TdT expression helps us identify a broader cluster with both lymphoid and myeloid potential, complementary to the current LMPP.

Reviewer #3 (Remarks to the Author):

The authors assess heterogeneity within human CD34+ hematopoietic stem and progenitor cells (HSPCs) using a highly multiplexed single cell proteomic approach. Given the unclarity about early cell fate decisions in human HSPCs, this is a study of outstanding interest in the field of hematopoiesis/HSPC biology/immunology.

The data acquisition is very robust and well controlled. While very decent numbers of biological replicates are used, the assessment of controls for the mass cytometry approach as well its bioinformatic analysis is outside the area of expertise of this referee.

We thank the reviewer for their appreciation of this work.

To zoom into HSCPCs, the authors identify the proteomic signature of CD34- cells and report a bifurcation between cells with lymphoid and myeloid potential compared to erythroid and megakaryocytic potential and find the bifurcation confirmed in transcriptomic and epigenetic landscapes. This is of outstanding interest. This part of the study would very much benefit from validation of this claim through the establishment of a cell surface-based prospective isolation protocol of key populations complemented by functional validation of this claim. Such validation could be run through comparable assays as the authors have used for TdT+ GMPs in the manuscript.

We agree with the reviewer that the Ly-My versus Ery-Mk bifurcation at the proteomic level is an exciting finding. We have included the differential protein expression analysis results for all pairwise combinations among meta-clusters in Supplementary Table 2. From there, we could identify 8 protein targets that were statistically significantly differed (adjusted p-value <0.05) in their expression between Meta-clusters A2 Erythroid progenitor and A4 Early lympho-myeloid (Ly-My) progenitors from our proteomic screen. This could be candidate markers to elucidate the early Ly-My vs. Ery-Mk bifurcation.

Part of Supplemental Figure 2]

marker	gate1	gate2	pvalue	avg	fc	comparison
CD34	A4	A2	1.27E-04	1.54706949	0.11697785	A4:A2
CLA	A4	A2	0.02131881	1.02331126	2.34721938	A4:A2
CD38	A4	A2	0	0.51835011	-0.4247686	A4:A2
SATB1	A4	A2	0	0.48678594	2.11631602	A4:A2
CD71	A4	A2	0	0.35251843	-4.1196267	A4:A2
CD45RA	A4	A2	1.04E-05	0.12498494	1.14720688	A4:A2

However, the functional follow-up and validation of this separate progenitor branch will require extensive work and an entirely different set of assays that are likely worth another entire manuscript. Thus, we have chosen to keep the manuscript focused on the lympho-myeloid axis here – but we really do appreciate the reviewer's enthusiasm here!

We would like to point the review towards another recent preprint from our lab that addresses this Erythroid/Myeloid axis and direct differentiation trajectory in more detail as part of a broader study of CD34 HSPCs spanning hematopoietic tissues (Favaro and Glass et al., biorxiv 2023 - <https://www.biorxiv.org/content/10.1101/2023.08.30.555623v1>).

Based on protein expression patterns, the authors identify a TdT+ cell cluster that they allocate between CD38lo LMPPs and CD10+ progenitors. Mapping of sc-proteome data to canonical HSPC

identification revealed unanticipated heterogeneity within the lympho-myeloid differentiation axis with clear bimodal division of the canonical GMP population into TdT+ vs TdT- cells. Immunophenotypical TdT+ GMPs are termed 'previously unappreciated lymphoid progenitors', based on a lymphoid-primed chromatin accessibility landscape that was determined using self-generated and available data sets. Next, the authors show that the expression of CD84 inversely correlates with TdT positive GMPs and enriches for putative lymphoid progenitors within GMPs. The authors confirm convincingly the distinct progenitor potential using suitable stromal layer-based and colony assays.

We appreciate the reviewer's summary and comments and hope the additional functional assays in the revision further strengthen the manuscript.

minors

This referee is surprised about the sharpness of the statement of the authors in the opening paragraph of the introduction on the concept of stepwise differentiation stages of hematopoietic progenitors. This is perceived as unnecessary by this referee because this was a very suitable model based on the methodologies available for a long time and it was interpreted very widely, thus, plasticity during differentiation stages was acknowledged and assumed, and therefore not treated as an 'ideal concept of progenitor cell types'. This referee recommends rephrasing, which has no effect on the importance of the current study.

We thank the reviewer for the feedback. We agree that the stepwise hematopoiesis concept was a very helpful, suitable model to understand the science and has been interpreted widely. We have modified that part of the manuscript.

[Revised manuscript line 27-37]

Our understanding of hematopoiesis has evolved dramatically in the last decade with the advances of single-cell technologies. Traditionally, hematopoiesis has been portrayed as a hierarchical system in which hematopoietic stem cells (HSCs) differentiate into oligo-potential and uni-potential progenitors in a stepwise manner. And each progenitor population with a distinct differentiation potential was identified by their expression of specific cell surface proteins, also known as 'surface markers' (Table 1). However, single-cell techniques revealed the continuous transcriptomic¹, epigenetic^{2,3}, and proteomic⁴ landscapes of human hematopoietic stem and progenitor cells (HSPCs). In parallel, single-cell level differentiation studies demonstrated that cells within the same 'population' based on the surface markers exhibit heterogeneous differentiation potentials^{5,6}. Thus, the initial surface phenotype definitions of HSPC cell types need updates to reflect recent findings.

The authors are asked to include information on the specificity of the antibodies used in each dot plot of Fig S7a.

We thank the reviewer for the comment. The antibody info is updated in Table S3, and the plots of Fig S7A are updated to include the antibody info.

[Part of Revised Supplemental Table 3]

Sorting Panel for all in vitro differentiation assays (Figure 6, Figure S7A)							
Laser	Ab	Clone	Dye	Vendor	Primary (per 100µl)	2ndary (per 100µl)	
Blue 488	CD34	8G12	PerCP/Cy5.5	BD	8		
Violet 405	CD38	E17-1519	BV421	Biolegend	4		
	CD45RA	HI100	BV605	Biolegend	4		
	Live/Dead		Aqua	Thermo Fisher			
	Dump (streptavidin)		BV510	Biolegend		5	
	cd3	UCHT1	biotin	Biolegend	2		
	cd11b	M1/70	biotin	Biolegend	2		
	cd14	M5E2	biotin	Biolegend	2		
	cd20	2H7	biotin	Biolegend	2		
	cd56	5.1H11	biotin	Biolegend	2		
	cd61	Y2/51	biotin	Biolegend	10		
	cd66b	B1.1/CD66	biotin	BD	1		
	cd235a	HIR2	biotin	Biolegend	2		
	UV 355	CD123	6H6	BUV737	BD	2	
	Red 640	CD84	CD84.1.21	APC	Biolegend	4	
Yellow 561	CD10	HI10a	PE-Cy7	Biolegend	2		

[Revised Supplemental Figure 7A]

Reviewer #4 (Remarks to the Author):

The study submitted by Kim et al. is an exciting tour-de-force, using a high throughput proteomic screen of human bone marrow progenitors to identify a subset of TdT+ GMPs with mixed lymphoid and myeloid potential, with these cells enriched in the CD84-low subset of GMPs. These data are therefore important for the field, elucidating new complexities in lympho-myeloid differentiation. Furthermore, these studies are performed with human progenitors and identify cell surface molecules which can be used to isolate live cells.

We thank the reviewer for the positive summary and the comments.

Specific comments:

I may have missed this information but it wasn't clear to this reviewer as to the percentages of cells that encompass each cluster from A1 to A10.

We thank the reviewer for the feedbacks. We have updated supplemental figure 1 to include the percentage of cells in each cluster and attached the panel below.

[Revised Supplemental Figure 1F]

F Distribution of cells across Meta-clusters

It might also be interesting to present the cluster map as a function of the cell cycle phase.

While we do not exactly have measurements of which cell cycle phase each cell is in, we have measured Ki67 level of each meta cluster (in supplemental figure 2E, which is attached below), that can be interpreted as the percentage of cycling cells. We have less than 5% of cells cycling in earliest A1 meta-cluster, and about 20% of cells in later clusters, A7, A9, and A10.

[Revised Supplemental Figure 2E]

From the data in Fig 5C/D, it appears that approximately 1% of all CD34+ cells are TdT-high GMPs or 8% of total GMPs. Is this correct? This information could be presented in the manuscript (and compared with information from previous publications in lines 403-404).

In the revised manuscript, we added a panel Fig S3E to describe the frequency of TdT+ GMP (average 3% of all CD34+ HSPCs) in bone marrow. While the frequency of TdT+ GMP seemed consistent across donors, we noticed the percentage of TdT+ GMPs over total GMPs varied,

depending on the GMP gate per donor. We have attached the updated panel and the manuscript section below.

[Revised manuscript lines 222-227]

We further gathered additional BM CD34+ HSPC data that were collected on CyTOF from Favaro and Glass et al. (bioRxiv, 2023) and quantified the frequency of TdT+ GMPs among all CD34+ HSPCs. TdT+ GMPs comprised 3.20% (standard deviation 1.00%) of total CD34+ HSPCs. This frequency was significantly higher than the frequency of LMPPs (mean frequency: 0.28%, paired t-test p-value: 1.06×10^{-5}) and CLPs (mean frequency: 1.82%, paired t-test p-value: 0.031) (Figure S3E).

[Revised Supplemental Figure 3E]

From a technical perspective, as relates to diversity between samples, the authors indicate that they used BM from 8 healthy adult donors. It would be helpful to indicate the age range of these donors. Additionally, in some of the figures (i.e. fig 2, 3, 5), it wasn't clear as to how many BM donors were used for the analyses. While outside the scope of this study, it would be of interest to assess whether these clusters are present at different levels in young vs old donors (as well as CD84-Hi vs CD84-Lo GMP). The authors may want to comment on this in the discussion.

We thank the reviewer for the comment. We have added the donor age and gender information as Supplemental Table 4 and updated the figure legends to clarify which donors were used for which experiments. Donor 9 was added during the revision for additional experiments. We do agree the change of HSPC composition throughout aging is of great interest, but within this manuscript, the number of donors used in each experiment was too low (max 3) to draw a confident conclusion. We anticipate future studies to investigate the effects of aging in hematopoiesis, leveraging lymphoid progenitor cell type identities from this manuscript.

[Revised Supplemental Table 4]

Donor #	Vendor	Gender	Age
1	AllCells	F	47
2	AllCells	F	34
3	AllCells	M	54
4	Stemexpress	F	47
5	AllCells	F	29
6	AllCells	M	50
7	AllCells	F	25
8	AllCells	M	51
9	AllCells	F	25

The identification of a TdT+ GMP is a very interesting and novel aspect of this manuscript. In this regard, what is the half-life of TdT? If the half-life is very long, this might result in maintained expression.

Previous study by Bentolila et al. (EMBO J. 1995) showed that two splicing isoforms, short TdT (TdTS) and long TdT (TdT_L), exist in mice, and the half-life of each isoform was 14h and 6-8h, respectively. While we were not able to find the corresponding study in humans, we imagine the half-life to be of a similar range. Thus, we expect TdT to be diluted rapidly upon cell cycle, which in human typically take about 24h in proliferating cells. Therefore, we would assume that TdT detected in BM HSPCs is reflective of current lymphoid-primed state. Furthermore, a recent publication utilizing and endogenous TdT in mice as a reporter (Klein et al., Nature Immunology, 2022) showed that TdT expression is not detected in mice GMP. Hence, TdT expression in human GMP is unlikely to be the remnant TdT from earlier progenitor, but more likely to be the evidence of an ongoing lymphoid gene regulatory program and the evidence of an imperfect identification scheme of the human GMP.

That being said, the data in Fig. 3B show levels of TdT in GMP that are equivalent to that in CLP. Are mRNA levels similar in these two subsets ((ie. CD84-low GMP vs CLP)?

We have analyzed publicly available index-sorted single-cell RNA-seq study from Velten et al. (Nature Cell Biology, 2017), using their web-based interface of the dataset. In the plot below, each dot represents a pre-gated CD38+CD45RA+ cell, which only consists of CLPs and GMPs, and the color of the dot is TdT (gene name DNTT) transcript level. Consistent with the protein level, a fraction of GMPs have DNTT transcripts as high as CLPs. Thus, the TdT expressing GMP population can be similarly identified at mRNA level.

The authors state that they focused on T lymphoid potentials because “B lymphoid cell origins in humans have been well documented (line 374).” Nonetheless, it is interesting to note that a murine progenitor population in the thymus with T and myeloid potential, but not B lymphoid potential, has been described by 2 groups (10.1038/nature06840, 10.1038/nature06839). As such, it would be interesting to determine whether CD84-Low GMP possess lymphoid activity and at the least, to discuss their data in comparison with this murine population.

In the revised manuscript we performed additional experiments and have measured the B cell

lineage potential, revised Figure 6D. We observed significant B cell potential from CD84^{lo} GMPs, but not as effective as CLPs. However, we do note that the *in vitro* B cell differentiation condition cannot be identical to *in vivo* potentials, and we anticipate future studies utilizing single-cell barcoding techniques to reveal the clonal relationship between human bone marrow lymphoid progenitors and the B-T lineages.

[Revised Figure 6D]

In the updated manuscript, we have discussed our data in comparison with murine populations.

[Revised manuscript lines 407-416]

With *in vitro* differentiation assays, we have shown all lymphoid lineage potentials – T, B, and NK – of the CD84^{lo} population. Especially for T cell potential, CD84^{lo} population exhibited ~17 times higher frequency of T cell progenitor than that of CD10⁺ CLPs (Figure 6E), while CD10⁺ CLPs exhibit more efficient B cell differentiation (Figure 6D). Thus, CD84^{lo} cells and CD10⁺ cells in humans strongly resemble Ly6D⁻ all lymphoid progenitors (ALPs) and Ly6D⁺ B-cell-biased progenitor (BLP) in mice²⁰. At the same time, the myeloid potential of CD84^{lo} population could be compared to the human LMPPs or murine MPP4s⁵¹. Based on the high dimensional proteomic and epigenomic analyses, we might postulate that TdT⁺ CD84^{lo} and TdT⁻ CD84^{lo} cells each resemble murine ALPs and MPP4s.

Minor

The authors state that they “identified CD84^{lo} as a surrogate for TdT⁺ GMPs for live-cell sorting (line 341).” While their sorting resulted in an impressive 40% of TdT⁺ cells, this reflects an important enrichment but not an identification. Conversely though, the authors clearly show that TdT⁺ GMP are not present in the CD84-high subset. The wording of this phrase can be modified.

We appreciate the reviewer for the comment. We revised the wording of the sentence to reflect the feedback.

[Revised manuscript lines 368--371]

Furthermore, we identified CD84^{lo} as the surface phenotype to enrich TdT⁺ GMPs for live-cell sorting, where CD84^{lo} GMPs yield robust lymphoid output in cellular differentiation assays. In contrast, CD84^{hi} GMPs were nearly devoid of TdT⁺ GMPs and showed lack of lymphoid potentials.

Reviewer #5 (Remarks to the Author):

Kim et al use high dimensional immune profiling to identify TdT- CD84lo granulocyte-monocyte progenitors (GMP) as upstream of lympho-myeloid progenitors. Altogether this is a very rigorous piece of work with sound conclusions.

We thank the reviewer for their summary and positive comments.

Minor:

1. Please provide information on the number events and, or range acquired and comment regarding confidence in identifying specific subsets.

We appreciate the reviewer's feedback on the details of the sample acquisitions. The number of events acquired for all experiments are updated in the revised manuscript.

For example, Figure 3A's legend has been updated to include total CD34+ cell numbers and Figure 3A has been updated to include the percentage of each population.

[Revised Figure 3A]

(A) Gating scheme of conventional HSPC cell types on CyTOF. Pregate: Singlet Live Lin- CD45lo CD34+ cells (n=10,167) from Figure 2D.

The number of cells collected for inTAC-seq has also been updated in the revised manuscript.

[Revised manuscript lines 616-617]

Donor 5: TdT+ GMP 1,500 cells, TdT- GMP 14,000 cells, Donor 6: TdT+ GMP 2,300 cells, TdT- GMPS, 15,000 cells

For experiments where the number of cells in the input or the collection of the data was assigned by the protocol, the methods section includes the details.

In general, we collected >100 events for a subset to be confident about the subset identification. For TdT+/TdT- or CD84lo / CD84hi subset, we defined the subset based on the bimodal peaks in this histogram and left the intermediate population between the peaks out from the analyses.

2. I found the figures legends too brief. I was unable to interpret the figures independent of the main text.

We thank the reviewer for the comment. We strive to allow such interpretation. We have updated the figure legends to be more thorough, with relevant experimental information throughout the revised manuscript. An example is shown below with changes tracked in the revised manuscript document.

Example – Figure 1A legend. Changes are highlighted in yellow.

Figure 1. Single-cell proteomic map of human bone marrow HSPCs

(A) Experimental overview of the single-cell proteomic screen.

(B) Heatmap of scaled median expression of molecules that were detected in at least 0.1% of CD34+ HSPC by meta-clusters. Columns and rows are hierarchically clustered. Meta-cluster annotation is in (C).

(C) Violin plots of the conserved core panel protein expressions by meta-clusters. Expression levels are normalized to the 99.9th percentile of each molecule.

(D) UMAP of all CD34+ cells (left) and of FlowSOM clusters (right). Cell-level UMAP was created with the conserved panel markers measured in each cell and FlowSOM cluster-level UMAP was created with median expression levels of all molecules of each cluster.

(E) UMAP of all CD34+ cells colored by IdU staining (left) or Integrin a9b1 staining (right).

(F) Percentage of positive cells expressing protein molecules that are increasing (left) or decreasing (right) along the hematopoietic differentiation.

3. The main text is very heavy reading. The work would benefit from a few explanatory sentences to help readers who are less expert in high dimensional analysis. Similarly, there are many long complex sentences that could be split to make the text more intelligible

We thank the reviewer for the feedback. We understand that our manuscript utilizes many different high-dimensional analyses, which the broader audience might not be familiar with. We have included detailed description of each analysis in the methods.

For example, we have a section on the detailed steps of the screening data analyses.

[Revised manuscript lines 669-675]

Proteomic screen data integration and clustering

To integrate the data collected on 15 different panels, we first clustered cells into 400 FlowSOM clusters⁵⁹ (median cell number per cluster: 1288) using the conserved panel. Then each FlowSOM cluster was given the median value for each target in the screen. For meta-clustering, we generated a nearest-neighbor graph of FlowSOM clusters using all median values of 81 targets and then performed Leiden-clustering using FindNeighbors and FindClusters functions, respectively, from Seurat package (<https://satijalab.org/seurat>).

We have made best efforts to revise the manuscript to make it more approachable. Furthermore, if the manuscript is accepted, we will work with the editorial team to make sure the text is appropriate for a general audience.

REVIEWER COMMENTS

Reviewer #1 (Remarks to the Author):

The revised version of this paper is somewhat improved. It does, however, not address all my concerns and the new data raise some additional issues.

My first specific point in the former version was that “I do not believe that the data support the title as no functional link between TdT expression and lymphoid lineage potential is established. The authors only examine the lineage potential of CD84^{low} cells among which only about one third are TdT positive and the molecular analysis does not make a strong case for lymphoid priming.”

Despite an extensive response, I cannot see that this address the concern that the authors use CD84 and not TdT for their analysis. Hence, I do not find that the title is supported by the data.

My former point two “The authors use the term lymphoid potential in the title but they only examine T/NK-lineage potential. Determination of the B-lineage potential of both CD84^{low} and CD84^{high} cells would provide more conclusive evidence to this end.”

While this has been addressed, it is unclear to me how the experiment is designed. I cannot see that the materials and methods section or the reporting summary has been updated with information about the B-cell differentiation assay.

The authors has done a single cell cloning assay with interesting data. However, the text is written as if the experiment has only been done once from a single donor.

Looking over the manuscript it would appear that the number of experiments and donor samples analyzed in each experiment are not indicated in the figure legends or M&M.

Material and methods mentions 9 donors but apart from that I found most limited information. In figure 4C, 5D it is indicated that the data are collected from 2 donors but otherwise information is most limited. I find this especially concerning for the functional analysis in figure 6.

Reviewer #2 (Remarks to the Author):

The authors replied to most concerns from this reviewer.

1. There remains one unanswered question, the surrogate nature of CD84 low expression.

The value of CD84low expression as a surrogate for TdT expression should be tuned down.

2. It was surprising that in the B cell assay LMPP did not yield CD19 cells. This result begs some clarification.

3. In the mouse, adult progenitors that do not have a myeloid potential in vivo yield myeloid cells in an OP9dll4 co-culture (Schlenner et al Immunity 2010). This observation raises some questions on the single cell analysis to evaluate the differentiation potential of TdT+ CD84low cells. Are CLP in the same assay giving rise to CD14+ or CD15+ cells?

Reviewer #3 (Remarks to the Author):

The authors have addressed all points of concern raised by this referee.

Reviewer #4 (Remarks to the Author):

I thank the Reviewers for their careful responses to the issues raised.

Reviewer #5 (Remarks to the Author):

The authors have been very responsive to reviewer feedback. The manuscript is much improved. I have no further queries.

REVIEWER COMMENTS

Reviewer #1 (Remarks to the Author):

The revised version of this paper is somewhat improved. It does, however, not address all my concerns and the new data raise some additional issues.

My first specific point in the former version was that “I do not believe that the data support the title as no functional link between TdT expression and lymphoid lineage potential is established. The authors only examine the lineage potential of CD84^{low} cells among which only about one third are TdT positive and the molecular analysis does not make a strong case for lymphoid priming.”

Despite an extensive response, I cannot see that this address the concern that the authors use CD84 and not TdT for their analysis. Hence, I do not find that the title is supported by the data.

We thank the reviewer for the comment. We have updated the title and main text to further explain the relationship between TdT and CD84^{lo} phenotype in the lymphoid progenitors.

New title:

Terminal deoxynucleotidyl transferase and CD84 identify human multi-potent lymphoid progenitors

Revised manuscript lines 370-377

Although previously regarded as myeloid committed within the canonical HSPC classification, TdT⁺ GMPs exhibit lymphoid bias in both proteomic and epigenetic landscapes that are distinct from the rest of TdT⁻ GMPs. Furthermore, we identified CD84^{lo} as the surface phenotype to enrich TdT⁺ GMPs for live-cell sorting, where CD84^{lo} GMPs yield robust lymphoid output in cellular differentiation assays. In contrast, CD84^{hi} GMPs were nearly devoid of TdT⁺ GMPs and showed lack of lymphoid potentials. Thus, we report strong molecular lymphoid bias in TdT⁺ GMPs and previously unappreciated lymphoid potentials in CD84^{lo} GMPs in the human bone marrow.

My former point two “The authors use the term lymphoid potential in the title but they only examine T/NK-lineage potential. Determination of the B-lineage potential of both CD84^{low} and CD84^{high} cells would provide more conclusive evidence to this end.”

While this has been addressed, it is unclear to me how the experiment is designed. I cannot see that the materials and methods section or the reporting summary has been updated with information about the B-cell differentiation assay.

We appreciate the reviewer’s comment and updated the methods section on B-cell differentiation assay. Our experiment used the culture condition from Scheeren F.A. et al.,

Eur. J. Immunol., 2010, with the input populations being LMPPs, CLPs, CD84lo GMPs and CD84hi GMPs, consistent with other experiments in this manuscript.

Revised manuscript lines 655-663

OP9 bulk co-culture differentiation assay for B cell differentiation

1 confluent 100mm OP9 plate was divided into 2 6-well plates and incubated overnight in a 37°C, 5% CO₂ incubator. 24hr after plating OP9 cells, 500 cells from each sorted HSPC population were deposited into each well and were cultured in the presence of 10 ng/ml IL-7 (all from Peprotech, London, UK) in OP9 media (SF7 media). Cells were dissociated from wells by pipetting, filtered with 70µm cell strainer, centrifuged at 300g for 5 minutes and transferred to new plates with fresh OP9 after a week. After 2 weeks, cells were harvested and analyzed by flow cytometry (Flow panel in Table S3). Entire volume in each well was analyzed and wells with fewer than 5 human cells (live hCD45+) were excluded from analysis.

The authors has done a single cell cloning assay with interesting data. However, the text is written as if the experiment has only been done once from a single donor.

We have also added replicates for the single-cell cloning assay in Figure 6G. With more replicates, we see consistent bipotentiality form single-cell differentiation. Thus, we suggest that the CD84lo cells are multipotential lymphoid progenitors.

Revised manuscript Lines 352-357

Out of 312 wells cloned, 18.6% (58 wells / 312 wells) were successfully cloned. Of the wells positively cloned, 43.4% (standard deviation = 10.0%) had both lymphoid and myeloid progeny, suggesting a substantial lympho-myeloid bipotentiality at clonal level among CD84lo cells. While the lymphoid-only or myeloid-only frequency showed a large donor-to-donor variation, the bipotentiality was observed consistently across three replicates (Figure 6G).

Revised Figure 6G

G Single-cell differentiation

Looking over the manuscript it would appear that the number of experiments and donor samples analyzed in each experiment are not indicated in the figure legends or M&M. Material and methods mentions 9 donors but apart from that I found most limited information. In figure 4C, 5D it is indicated that the data are collected from 2 donors but otherwise information is most limited. I find this especially concerning for the functional analysis in figure 6.

We thank the reviewer for the comment. We have updated the figure legends and sections of Material and Methods as below to articulate the number of donors analyzed in each section.

Revised manuscript lines 526-527

CyTOF screen staining and acquisition (Workflow in Figure S1B)

For CyTOF screen, fresh mononuclear cells collected from three donors, each 50ml of BM, were used for experiments.

lines 577-580

CyTOF staining for single panels

For CyTOF experiment for HSPC panel validation (Figure 2D~Figure 3) or surface marker identification (Figure 5), 10×10^6 cells per donor were thawed from liquid nitrogen, using the thawing media described above.

lines 604-605

FACS sorting for inTAC-seq

For inTAC-seq, all mononuclear cells collected from two donors, each 50ml of BM, were used for experiments.

lines 616-617

FACS sorting for live cells

For FACS sorting for functional differentiation experiments (Figure 6), 10×10^6 cells per donor were thawed from liquid nitrogen, using the thawing media described above.

Legends

Figure 1. Single-cell proteomic map of human bone marrow HSPCs

(A) Experimental overview of the single-cell proteomic screen. Total three healthy human bone marrow (Donor 1~3) were analyzed for the proteomic screen.

Figure 2. Identification of the proteomic signatures of human BM HSPC populations

(D) Workflow for the follow-up CyTOF experiment. One additional bone marrow (Donor 4) was analyzed to validate the screened and selected protein targets. A different healthy donor bone marrow was used and CD34- cells were also included after density-downsampling.

Figure 4. TdT+ subset of human GMPs exhibit a lymphoid-primed chromatin accessibility landscape

(A) Workflow of inTAC-seq. **Two healthy human bone marrow (Donor 5~6) were sorted for TdT- and TdT+ GMPs.** GMPs were gated based on surface markers and then sorted into TdT+ or TdT- population by TdT intracellular staining.

Figure 6. CD84^{lo} GMPs yield robust multi-lymphoid output with in vitro differentiation assays

(A) Workflow of *in vitro* differentiation assays. **Bone marrow from Donor 7, 8 with two technical replicates per donor were used for bulk co-culture assays, limiting dilution assay, and methylcellulose assay and one sample of Donor 3 and two replicates from Donor 9 were used for single-cell cloning assay**

Reviewer #2 (Remarks to the Author):

The authors replied to most concerns from this reviewer.

1. There remains one unanswered question, the surrogate nature of CD84 low expression. The value of CD84^{low} expression as a surrogate for TdT expression should be tuned down.

We thank the reviewer for their comment. We have updated the title and main text to explain the relationship between TdT and CD84^{lo} phenotype in the lymphoid progenitors.

New title:

Terminal deoxynucleotidyl transferase and CD84 identify human multi-potent lymphoid progenitors

Revised manuscript lines 370-377

Although previously regarded as myeloid committed within the canonical HSPC classification, TdT⁺ GMPs exhibit lymphoid bias in both proteomic and epigenetic landscapes that are distinct from the rest of TdT⁻ GMPs. Furthermore, we identified CD84^{lo} as the surface phenotype to enrich TdT⁺ GMPs for live-cell sorting, where CD84^{lo} GMPs yield robust lymphoid output in cellular differentiation assays. In contrast, CD84^{hi} GMPs were nearly devoid of TdT⁺ GMPs and showed lack of lymphoid potentials. Thus, we report strong molecular lymphoid bias in TdT⁺ GMPs and previously unappreciated lymphoid potentials in CD84^{lo} GMPs in the human bone marrow.

2. It was surprising that in the B cell assay LMPP did not yield CD19 cells. This result begs some clarification.

We agree with the reviewer that LMPP not yielding CD19 cells could seem surprising. However, it has been reported that even HSCs are inefficient in B cell differentiation in vitro (Kondo et al., Cell, 1997). Especially for OP9 coculture system, it has also been reported that earlier progenitors cannot yield CD19⁺ B cells successfully unless FLT3L is supplemented (Cho et al., PNAS, 1999). Thus, we suspect the LMPPs are too early in B cell development pathway to differentiate in our OP9 coculture system with only IL-7 supplemented, which is sufficient for CD84^{lo} cells or CLPs. We have added this interpretation in our revised manuscript, as copied below.

Lines 318-325

We note that CD38^{lo} LMPPs did not yield CD19⁺ B cells in our culture condition with IL-7 only. It has been reported earlier progenitors require FLT3 ligand (FLT3L), while more downstream lymphoid progenitors which are already responsive to IL-7 do not require FLT3L⁵². Thus, we conclude that while CD38^{lo} LMPPs are the earliest lymphoid progenitors that are not yet responsive to IL-7, CD84^{lo} cells have robust lymphoid potential to all lymphoid lineages (T, B, NK) in vitro, and CD10⁺ CLPs are highly efficient B cell progenitors. On the other hand, CD84^{hi} myeloid progenitors are devoid of any lymphoid potentials.

3. In the mouse, adult progenitors that do not have a myeloid potential in vivo yield myeloid cells in an OP9dll4 co-culture (Schlenner et al Immunity 2010). This observation raises some questions on the single cell analysis to evaluate the differentiation potential of TdT+ CD84low cells. Are CLP in the same assay giving rise to CD14+ or CD15+ cells?

We thank the reviewer for their feedback and introducing an interesting observation in previous literature. In our culture condition, human CLPs did not yield CD14+ or CD15+ cells in OP9-DL4 coculture condition. Moreover, they were very in efficient in making granulocytic or macrophage colonies in CFU assay as well. Thus, unlike the murine mouse models, human CLPs seem to be devoid of myeloid potentials in vitro.

Revised manuscript Figure 6B & 6F

Reviewer #3 (Remarks to the Author):

The authors have addressed all points of concern raised by this referee.
We thank the reviewer for their feedback for the manuscript.

Reviewer #4 (Remarks to the Author):

I thank the Reviewers for their careful responses to the issues raised.
We thank the reviewer for their feedback for the manuscript.

Reviewer #5 (Remarks to the Author):

The authors have been very responsive to reviewer feedback. The manuscript is much improved. I have no further queries.
We thank the reviewer for their feedback for the manuscript.

REVIEWERS' COMMENTS

Reviewer #1 (Remarks to the Author):

My remaining concerns are handled in this new version of the manuscript. Thank you.

Reviewer #2 (Remarks to the Author):

The authors answered the concerns

REVIEWERS' COMMENTS

Reviewer #1 (Remarks to the Author):

My remaining concerns are handled in this new version of the manuscript. Thank you.

We thank the reviewer for their feedback for the manuscript.

Reviewer #2 (Remarks to the Author):

The authors answered the concerns

We thank the reviewer for their feedback for the manuscript.